# The ER membrane protein complex restricts mitophagy by controlling BNIP3 turnover

Jose M Delgado [iD][1], Logan Wallace Shepard[1], Sarah W Lamson[1], Samantha L Liu[1] & Christopher J Shoemaker [iD][1,2]✉

## Abstract

Lysosomal degradation of autophagy receptors is a common proxy for selective autophagy. However, we find that two established mitophagy receptors, BNIP3 and BNIP3L/NIX, are constitutively delivered to lysosomes in an autophagy-independent manner. This alternative lysosomal delivery of BNIP3 accounts for nearly all its lysosome-mediated degradation, even upon mitophagy induction. To identify how BNIP3, a tail-anchored protein in the outer mitochondrial membrane, is delivered to lysosomes, we performed a genome-wide CRISPR screen for factors influencing BNIP3 flux. This screen revealed both known modifiers of BNIP3 stability as well as a pronounced reliance on endolysosomal components, including the ER membrane protein complex (EMC). Importantly, the endolysosomal system and the ubiquitin–proteosome system regulated BNIP3 independently. Perturbation of either mechanism is sufficient to modulate BNIP3-associated mitophagy and affect underlying cellular physiology. More broadly, these findings extend recent models for tail-anchored protein quality control and install endosomal trafficking and lysosomal degradation in the canon of pathways that tightly regulate endogenous tail-anchored protein localization.

**Keywords** BNIP3; Mitophagy; EMC; Secretory Pathway; TA Protein
**Subject Categories** Autophagy & Cell Death; Cell Adhesion, Polarity & Cytoskeleton; Membranes & Trafficking

## Introduction

Autophagy is an intracellular degradative pathway that clears unwanted cytoplasmic components such as damaged or superfluous organelles (Kirkin, 2020). During autophagy, a unique double-membrane vesicle—the autophagosome—is generated around cargo. The completed autophagosome subsequently traffics to the lysosome where its content is degraded. The recognition and clearance of mitochondria by autophagy (hereafter mitophagy) are broadly implicated in aging, development, and disease

(Nguyen et al, 2016; Ng et al, 2021). Immense progress has been made toward understanding the canonical PINK1/Parkin-dependent mitophagy pathway (Nguyen et al, 2016). However, mitophagy can occur independently of this machinery (i.e., PINK1/Parkin-independent) where it is executed by less-understood mechanisms varying across cell type and physiological context (Ng et al, 2021).

BNIP3 and BNIP3L/NIX are paralogous membrane proteins found on the outer mitochondrial membrane (OMM) (Bellot et al, 2009; Chen et al, 1997). As mitophagy receptors, BNIP3 and NIX recruit key autophagy proteins, in particular the Atg8-family of proteins (LC3 and GABARAP families in humans), to the surface of targeted mitochondria (Zhu et al, 2013; Hanna et al, 2012; Rogov et al, 2014). Such interactions enforce cargo specificity by keeping the expanding autophagosomal membrane in close apposition to the targeted mitochondrion. The potency of these interactions is reflected in the observation that ectopic expression of BNIP3 or NIX is sufficient to induce selective mitophagy (Kim et al, 2021; Quinsay et al, 2010; Lee et al, 2011). Thus, the expression and/or activation of BNIP3 and NIX must be appropriately constrained in vivo to spatiotemporally restrict aberrant mitophagy induction. Early studies identified transcriptional regulation by hypoxia-inducible factor 1 (HIF-1) as a key facet of BNIP3 and NIX regulation (Bellot et al, 2009). Consistent with this model, both BNIP3 and NIX expression and associated mitophagy are potently induced upon hypoxia onset. Recently, multiple groups have extended this model, reporting that the ubiquitin–proteasome system (UPS) potently restricts BNIP3 and NIX levels to further curb mitophagy (Alsina et al, 2020; Thanh Nguyen-Dien et al, 2023; Cao et al, 2023; Zheng et al, 2022; Poole et al, 2021; Elcocks et al, 2023). In light of these concepts, it is important to develop a unified understanding of how steady-state levels of these mitophagy receptors are established and maintained, and how this regulation governs underlying cell physiology.

BNIP3 and NIX are targeted to the OMM by a single, C-terminal transmembrane domain (TMD) (Yasuda et al, 1998). This topology defines a diverse class of membrane proteins (~50 in yeast, >300 in humans) known as tail-anchor (TA) proteins, which rely exclusively on posttranslational insertion mechanisms (Beilharz et al, 2003; Kalbfleisch et al, 2007; Borgese et al, 2003). TA protein targeting poses a fundamental and innate challenge for cells. The hydrophobicity of a TA TMD is a primary determinant of

[1]Department of Biochemistry and Cell Biology, Geisel School of Medicine, Dartmouth College, Hanover, NH, USA. [2]Dartmouth Cancer Center, Lebanon, NH, USA.
✉E-mail: Christopher.J.Shoemaker@Dartmouth.edu

 

its localization, with mitochondrially targeted TMDs having a lower hydrophobicity, on average, than those targeted to the ER (Wattenberg et al, 2007; Beilharz et al, 2003). However, this relationship is not absolute. In the OMM, TA proteins are inserted via MTCH1/MTCH2, while mislocalized or aberrant TA proteins are extracted by ATAD1 (Msp1 in yeast) (Guna et al, 2022; Wohlever et al, 2017). In the ER membrane, TA proteins are inserted by either the "guided entry of TA proteins" (GET) pathway or the "ER membrane protein complex" (EMC), while mislocalized or aberrant TA proteins are extracted by ATP13A1 (Spf1 in yeast) (Hegde and Keenan, 2011; Guna et al, 2018; McKenna et al, 2022; Wang et al, 2014). Far from futile, dynamic cycles of TA protein insertion and extraction play a critical role in properly partitioning TA proteins despite limited and overlapping targeting information (Hansen et al, 2018; McKenna et al, 2020; Xiao et al, 2021; Matsumoto and Endo, 2022; Matsumoto et al, 2019; McKenna et al, 2022). As representative TA proteins, BNIP3 and NIX are primarily localized to the OMM but have been demonstrated to localize to other membranes (Bozi et al, 2018; Hanna et al, 2012; Kanekura et al, 2015; Zhang et al, 2009; Wilhelm et al, 2022). Consequently, exploration of BNIP3 and NIX regulation has the potential to reveal additional insights into TA protein quality control mechanisms.

Here we utilized a triple-negative breast cancer cell line, MDA-MB-231, that forms dense hypoxic tumors in vivo to study the posttranslational regulation of BNIP3 in hypoxic and nonhypoxic conditions (Kim et al, 2018; Xie et al, 2016). We demonstrate a novel mode of BNIP3 degradation that is lysosome-mediated, but autophagy-independent, and accounts for most lysosome-mediated degradation of BNIP3. This pathway requires ER insertion by the EMC and subsequent trafficking through the canonical secretory pathway. Endolysosomal regulation works alongside, but independent of, UPS-mediated regulation of BNIP3, providing an additional regulatory axis for governing BNIP3-mediated mitophagy and its associated physiology. In the process, we directly implicate endosomal trafficking and lysosomal degradation in the canon of quality control pathways that ensure proper localization of TA membrane proteins.

## Results

### Lysosomal delivery of BNIP3 is independent of autophagy

Lysosomal degradation of autophagy receptors is a common proxy for selective autophagy. Using this rationale, we set out to monitor the lysosomal delivery of endogenous BNIP3. To this end, we used MDA-MB-231 cells, a triple-negative breast cancer cell line that prominently expresses BNIP3. As previously reported, BNIP3 appears as multiple bands via immunoblot, reflective of variably phosphorylated species, which we confirmed by an in vitro dephosphorylation assay (Fig. EV1A) (Poole et al, 2021; Kanekura et al, 2015). BNIP3 protein levels accumulated in MDA-MB-231 cells treated with Bafilomycin-A1 (Baf-A1), a V-ATPase inhibitor that blocks lysosomal acidification, confirming that BNIP3 is degraded in a lysosome-dependent manner (Fig. 1A). To test if this lysosomal delivery was mediated by autophagy, we transduced Cas9-expressing cells with a single-guide RNA (sgRNA) targeting *ATG9A*, a core autophagy component, and selected in puromycin for 8 days to generate a non-clonal knockout population.

Unexpectedly, the deletion of *ATG9A* did not affect BNIP3 protein levels or its response to Baf-A1 treatment. A similar trend was observed for the related mitophagy receptor, NIX. Importantly, canonical selective autophagy receptors p62 and NDP52 accumulated upon either Baf-A1 treatment or sgATG9A transduction as expected for bona fide autophagy substrates (Fig. 1A). Comparable results were obtained from a clonal $ATG9A^{KO}$ isolate (Fig. EV1B).

Because hypoxia induces BNIP3- and NIX-mediated mitophagy, we reasoned that autophagy-dependent lysosomal delivery of these factors might occur preferentially under hypoxic conditions. To test this, we incubated cells in low oxygen (1% $O_2$) for 18 h, whereupon we observed an increase in BNIP3 protein levels consistent with known transcriptional regulation (Fig. EV1C). Regardless, *ATG9A* still did not affect BNIP3 protein levels relative to control cells (Fig. EV1C). This autophagy-independent lysosomal degradation of BNIP3 was observed across a diverse panel of cell lines including U2OS, HEK293T, MDA-MB-435, and K562 (Fig. EV1D–F). From this, we conclude that BNIP3 (and to a lesser extent, NIX) constitutively undergo robust lysosomal-mediated degradation that is primarily independent of autophagy.

To better dissect the lysosomal delivery of BNIP3, we adapted a tandem fluorescent (tf) reporter composed of a red fluorescent protein (RFP) and a green fluorescent protein (GFP) fused to a protein of interest, in this case BNIP3 (Shoemaker et al, 2019; Kimura et al, 2007). GFP fluorescence is selectively quenched in the low pH environment of the lysosomal lumen. In contrast, RFP fluorescence persists (Fig. 1B). Therefore, the red:green ratio serves as a ratiometric proxy for lysosomal delivery and can be quantified with single-cell resolution. Utilizing the tf-reporter system, we generated Cas9-expressing MDA-MB-231 cells stably co-expressing N-terminally tagged BNIP3 from the AAVS1 safe-harbor locus. By this approach, we observed a striking collapse in the red:green ratio of our tf-BNIP3 reporter in cells treated with Baf-A1, consistent with our earlier observations (Fig. 1C). In contrast, inhibiting autophagy with a chemical inhibitor of VPS34, PIK-III, failed to collapse the red:green ratio of tf-BNIP3, despite inhibiting flux of a canonical autophagy reporter, tf-NDP52 (Fig. EV1G) (Ohnstad et al, 2020). By a complementary genetic approach, we similarly found that knockdown of RAB7A, a small GTPase broadly associated with the late endosomal system, collapsed the red:green ratio of tf-BNIP3 (Fig. EV1H), while tf-BNIP3 flux persisted cells lacking key autophagy-specific factors: ATG9A, FIP200, or ATG7 (Fig. 1C). To further validate the tf-BNIP3 reporter, we monitored tf-BNIP3 expression and localization in MDA-MB-231 cells using fluorescence-based confocal microscopy. In control cells, GFP signal strongly correlated with mitoBFP, evidence that tf-BNIP3 localizes appropriately to mitochondria (Fig. 1D, E) (Friedman et al, 2011). RFP-only puncta were prevalent in DMSO-treated controls but fully collapsed into RFP$^+$/GFP$^+$ puncta upon Baf-A1 treatment (Fig. 1D,E). These RFP-only puncta co-localized with a lysosomal marker, LAMP1, consistent with the interpretation that RFP-only structures reflect lysosomal delivery of the reporter (Fig. 1F). Similar results were observed in $ATG9A^{KO}$ cells, reinforcing that this process is autophagy-independent. As an aside, we note that Baf-A1 treatment depleted the correlation coefficient of RFP or GFP with mitoBFP, suggesting that lysosomally destined RFP$^+$/GFP$^+$ structures (i.e., BNIP3) do not contain luminal mitochondrial content (Figs. 1E and EV1I). Collectively, these data indicate that our tf-BNIP3 reporter

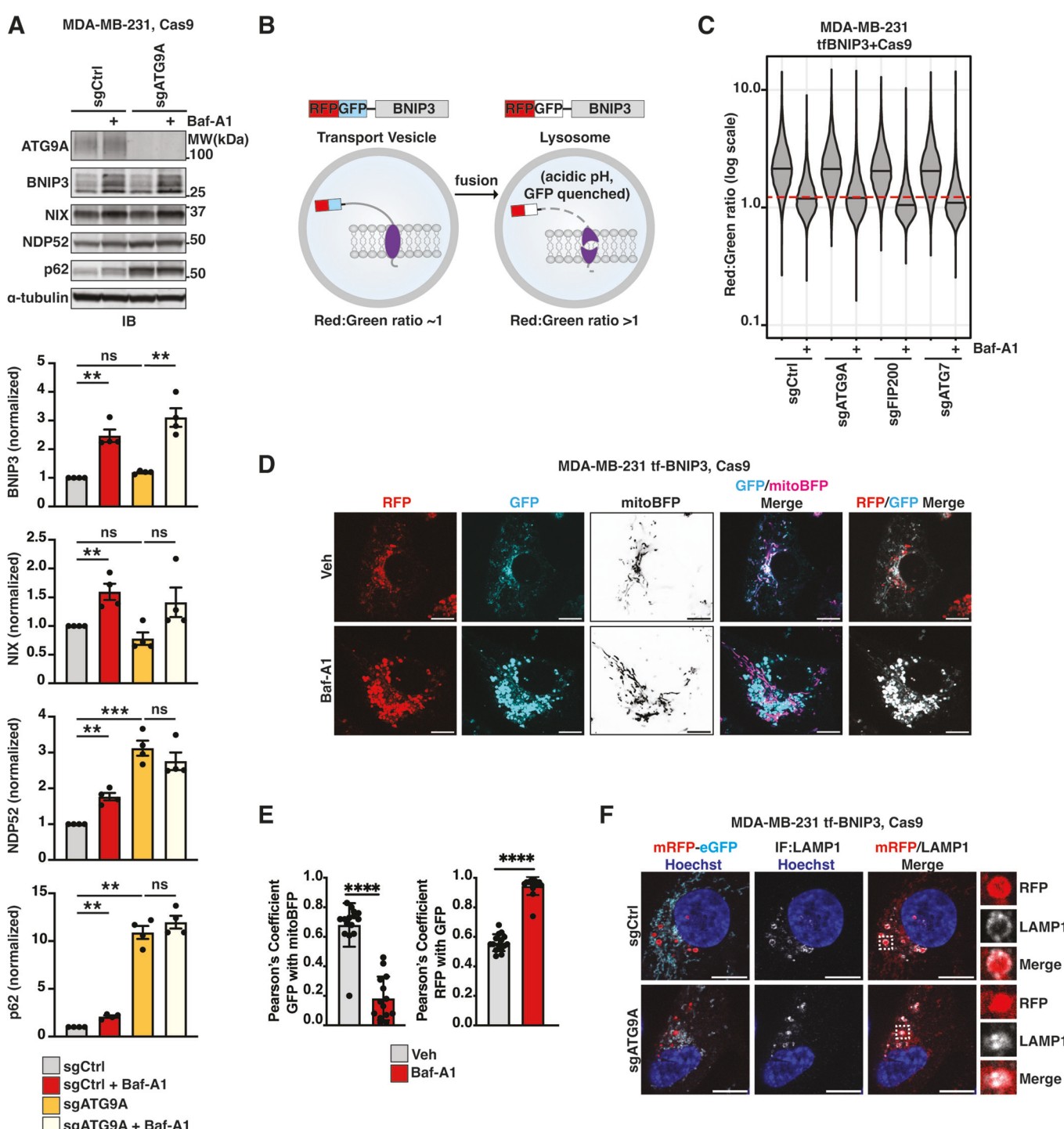

recapitulates the autophagy-independent degradation of BNIP3 by the lysosome.

## Genome-wide CRISPR screening reveals modifiers of BNIP3 flux

In the absence of an autophagy-mediated pathway, it was uncertain how an OMM protein would be robustly degraded by the lysosome. To identify factors required for the lysosomal delivery of BNIP3, we

employed our tf-BNIP3 reporter to perform a genome-wide CRISPR knockout screen for modifiers of BNIP3 flux. MDA-MB-231 cells expressing Cas9 and tf-BNIP3 were transduced with a lentiviral library containing 76,441 sgRNAs spanning the entire human genome (Doench et al, 2016) (Fig. 2A). Cells were then sorted by red:green ratio to collect the top and bottom 30% of cells, representing cells that were enhanced and inhibited for lysosomal delivery of tf-BNIP3, respectively (Fig. 2A). To identify genes associated with each effect, sgRNAs from each pool were amplified, sequenced, and analyzed with

**Figure 1.** Lysosomal delivery of BNIP3 is independent of autophagy.

(A) Immunoblotting (IB) of MDA-MB-231-derived extracts from cells expressing Cas9 and the indicated sgRNA. Where indicated, cells were treated with 100 nM Baf-A1 for 18 h. Shown are representative images from one biological replicate. Bar graphs represent mean $+/-$ SEM from four independent experiments. All protein levels were normalized to α-tubulin. Statistical analysis was performed using a one-sample $t$ test to the normalized control and an unpaired Student's $t$ test between experimental samples. Ctrl nontargeting control. ***$P$ <0.001; **$P$ <0.01; ns not significant. (B) Schematic of the tf-BNIP3 reporter. Upon lysosomal delivery, GFP fluorescence is selectively quenched. Thus, corresponding changes in red:green ratio reflect delivery to lysosomes. (C) tf-BNIP3-expressing cells were transduced with the indicated sgRNAs. Cells were subsequently treated with DMSO or Baf-A1 (100 nM) for 18 h before being analyzed by flow cytometry for red:green ratio. Median values for each sample are identified by a black line within each violin. The red dotted line across all samples corresponds to red:green ratio of maximally inhibited conditions (Baf-A1) (n >10,000 cells). (D) Representative confocal micrographs of tf-BNIP3 cells transiently expressing mitoBFP. Cells were treated with vehicle (DMSO) or Baf-A1 (100 nM) for 18 h prior to imaging. Scale bar: 10 μm. (E) Quantification of Pearson's correlation coefficients from cells in (D). Correlation of RFP with GFP (an anti-correlate of lysosomal delivery) and GFP to mitoBFP (reflective of mitochondrial localization) was calculated using Coloc2. Bar graphs represent mean $+/-$ SEM. Each data point represents a single cell. $n = 15$ cells. Statistical analysis was performed using an unpaired $t$ test. ****$P$<0.0001. (F) Representative confocal micrographs of cells transduced with sgRNA constructs targeting $ATG9A$ or a nontargeting control (Ctrl). Cells were fixed 8 days post transduction and immunostained for LAMP1 prior to imaging. Scale bar: 10 μm. Source data are available online for this figure.

the Model-based Analysis of Genome-wide CRISPR-Cas9 Knockout (MAGeCK) pipeline (Li et al, 2014; Martin, 2011; Deangelis et al, 1995) (see Dataset EV1). We utilized $\log_2$ fold change as a proxy for the strength of a gene as an effector of lysosomal delivery. A negative fold change indicates the gene mediates lysosomal delivery, as the perturbation leads to decreased flux. A positive fold change identifies genes that, when knocked out, induce flux. We categorized the two populations as potential "effectors" and "suppressors", respectively.

Any gene with a $\log_2$ fold change less than $-0.5$ or greater than 0.5 was considered a "hit" in the screen. At this threshold, we identified 122 effector genes and 112 suppressor genes (Fig. 2B; Dataset EV1). Concordant with our preliminary observations, core autophagy factors were absent from the effector population. Yet, we recovered RAB7A as an effector, as previously validated (Fig. EV1H). In addition, we recovered multiple known suppressors of BNIP3 flux, including mitochondrial protein import factors DNAJA3 and DNAJA11, the mitochondrial chaperone HSPA9 and, as recently reported, the outer membrane protein TMEM11 (Michaelis et al, 2022; Gok et al, 2023). In all, our list of identified effector and suppressor proteins was largely concordant with available data, validating our approach.

Surprisingly, when compared to the MitoCarta 3.0 database (Rath et al, 2021), only 1 of 122 effector genes and 21 of 112 suppressor genes were annotated as mitochondrial (Fig. 2C). To identify other pathways or components implicated by our data, we performed an unbiased Gene Ontology (GO) analysis. Enriched GO terms in the effector population related to membrane insertion, vesicle-mediated transport, and proteasomal pathways, with many terms specifically pertaining to the endoplasmic reticulum (ER) (Fig. 2D,E). Previously profiled autophagy receptors do not similarly enrich for these GO terms, suggesting a uniqueness to BNIP3 (Shoemaker et al, 2019). In particular, our data identified the EMC, the guided entry of tail-anchored proteins (GET) complex, ER-Golgi transport, and the ubiquitin–proteasome system (UPS) as potential effectors of BNIP3 stability (Fig. 2B). In sum, our genetic screening approach identified numerous known regulators of BNIP3 as well as a unique role for ER insertion and ER-to-Golgi trafficking in BNIP3 regulation.

## Lysosomal delivery of BNIP3 is governed by the EMC and the secretory pathway

To validate our screen results, we transduced our tf-BNIP3 reporter cells with a representative subset of individual sgRNAs and

monitored corresponding changes in red:green ratio using flow cytometry. These data clearly verified the EMC as a potent effector of BNIP3 degradation as the deletion of EMC subunits mirrored the effect of Baf-A1 treatment (Figs. 3A and EV2A). In addition, knockout of the GET complex, components of the secretory pathway including multivesicular bodies (MVBs), UPS factors, and vacuolar ATPase subunits all decreased red:green ratio (Figs. 3A and EV2A,B). Similar effects were observed in U2OS osteosarcoma cells expressing tf-BNIP3 and Cas9, confirming that the effectors we identified are not strictly cell-type specific (Fig. EV2C).

Within the endolysosomal system implicated above, the EMC and GET complex are related ER insertion pathways for tail-anchored proteins (Wang et al, 2014; Yamamoto and Sakisaka, 2012; Schuldiner et al, 2008; Mariappan et al, 2011; Guna et al, 2018). Notably, BNIP3 was previously observed on the ER membrane and accumulates on the ER during stress conditions (Bozi et al, 2018; Hanna et al, 2012; Kanekura et al, 2015; Zhang et al, 2009). In $EMC3^{KO}$ cells, tf-BNIP3 displayed a striking decline in RFP-only puncta with a concomitant increase in the co-localization of BNIP3 with mitoBFP (Fig. 3B). Knockout of vesicular transport factors $USO1$ or $SAR1A$ failed to fully prevent lysosomal delivery of BNIP3. However, both exhibited a marked shift in BNIP3 localization to structures resembling the ER network (Fig. 3B). Taking advantage of these differences in localization, we performed an epistasis analysis through pairwise depletion of $EMC3$ and $USO1$. While knockout of $USO1$ shifted the distribution of BNIP3 primarily to an ER-like morphology, when combined with the knockdown of $EMC3$, BNIP3 shifted back to a primarily mitochondrial localization (Fig. 3C). This epistatic relationship suggests the EMC governs BNIP3 entry into the ER membrane, which precedes the role of USO1 and the secretory pathway in trafficking BNIP3 to the lysosome. Relatedly, a recent report identified MTCH1 and MTCH2 as protein insertases governing mitochondrial TA protein insertion into the OMM (Guna et al, 2022). Consistent with those findings, dual ablation of MTCH1 and MTCH2 resulted in the localization of BNIP3 to the ER membrane, which did not disrupt lysosomal delivery (Fig. EV2D,E). Taken together, these data suggest that the EMC governs BNIP3 entry into the ER, while MTCH1/2 governs entry into the OMM.

To test whether BNIP3 trafficking through the endolysosomal system was an artifact of overexpression, we transduced Cas9-expressing MDA-MB-231 cells with sgRNAs targeting GET4, EMC3, USO1, or SAR1A and selected under puromycin for 8 days. We then subjected these cells to normoxic or hypoxic conditions

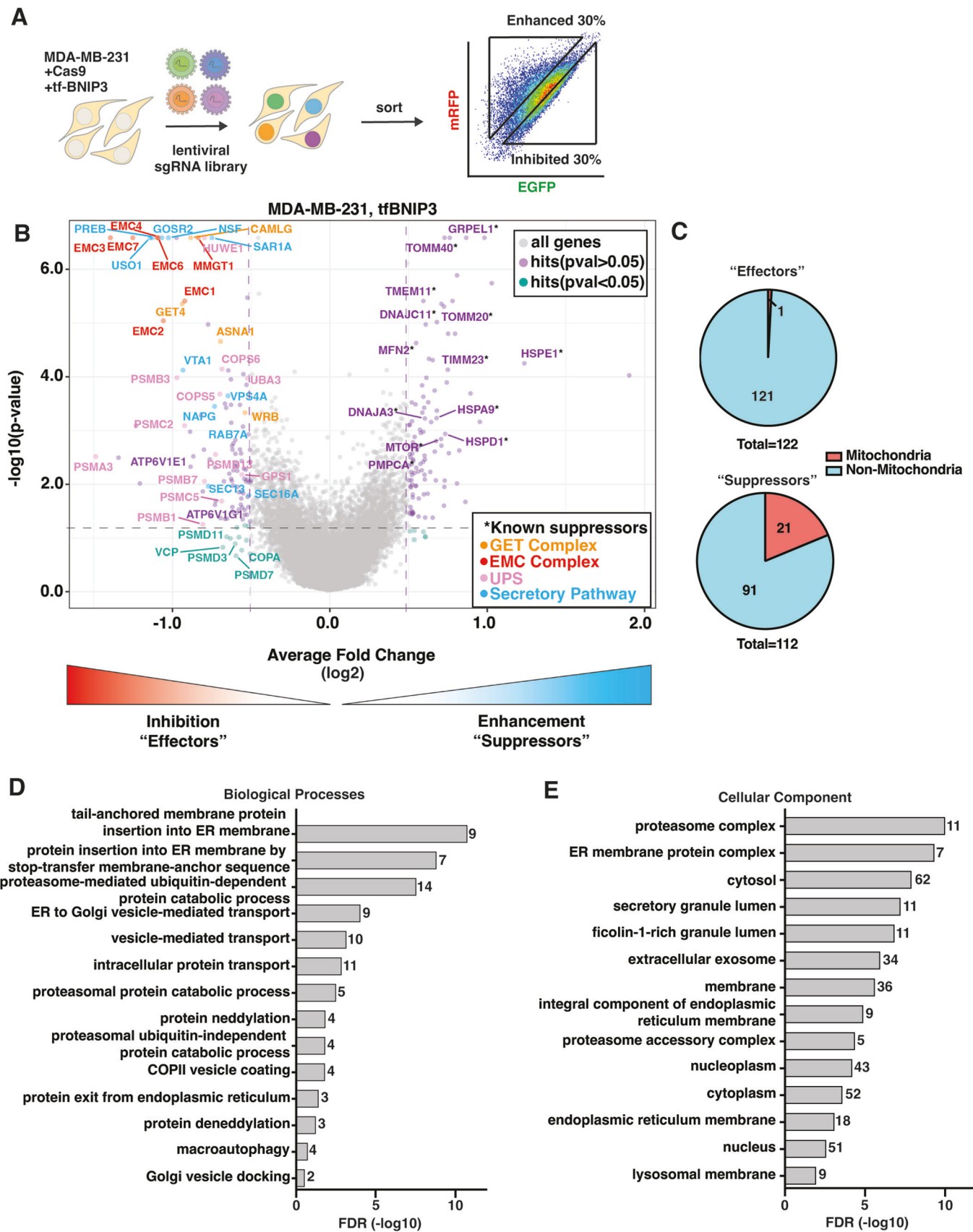

◀

**Figure 2. Genome-wide CRISPR screening reveals modifiers of BNIP3 flux.**

(A) Schematic of the genome-wide CRISPR screening pipeline for modifiers of tf-BNIP3 delivery to the lysosome. Reporter cells were transduced with an sgRNA library, propagated, and sorted to collect the top 30% (enhanced delivery) and bottom 30% (inhibited delivery) of tf-BNIP3 cells based on the red:green fluorescence ratio. (B) Volcano plot of BNIP3 effectors based on average fold change (a proxy for effect strength) and *P* value from $n = 3$ experimental replicates. Values were computed from robust rank aggregation (RRA) by the MAGeCK algorithm, a learned mean-variance model. Average $\log_2$ fold changes greater than 0.5 and less than −0.5 are indicated vertical dashed lines. The horizontal dashed line indicates a *P* value of 0.05. Only genes from cellular pathways or protein complexes validated by this study or independent studies are labeled. (C) Effectors and suppressors identified in (B) were mapped against the MitoCarta 3.0 database to identify known mitochondrial factors. (D, E) Unbiased Gene Ontology (GO) term analysis of genes in the effector population.

and monitored cellular extracts for changes in endogenous BNIP3 levels. All responses were measured in comparison to chemical inhibition of the lysosome by Baf-A1. When comparing each Baf-A1-treated knockout to its respective DMSO-treated control, we saw Baf-A1 sensitivity diminish (Fig. 3D). Knockout of *EMC3* remained the most potent effector, as BNIP3 protein levels were completely insensitive to Baf-A1 treatment in this background. Knockout of *GET4* and *USO1* resulted in a reduced sensitivity to Baf-A1, while knockout of *SAR1A* had only a minimal effect (Fig. 3D). Similar trends were observed in U2OS cells (Fig. EV2F). These results affirm that deletion of the EMC prevents lysosomal delivery of endogenous BNIP3. Strikingly, genetic perturbations that decreased sensitivity to Baf-A1 showed a concomitant increase in BNIP3 stabilization by bortezomib (BTZ), a potent inhibitor of the proteasome (Fig. EV2G). These data suggest that, upon inhibition of lysosomal delivery, the expected increase in total BNIP3 protein levels is attenuated by compensatory proteasomal degradation. We note that the BTZ-induced changes in the BNIP3 banding pattern reflect a combination of hyper-phosphorylated species and an accumulation of SDS-resistant dimers of BNIP3 (Fig. EV2G), formation of which we postulate is indirectly driven through the overall increase in BNIP3 levels (Poole et al, 2021).

As an independent measure of the role of the secretory pathway in delivering BNIP3 to lysosomes, we utilized a chemical inhibitor of ER-to-Golgi transport, Brefeldin-A (BFA). Treatment with BFA alone had no significant effect on steady-state levels of BNIP3. However, BFA treatment fully negated the stabilizing effects of Baf-A1, consistent with the model that ER-to-Golgi trafficking of BNIP3 is a prerequisite for its lysosomal delivery. In contrast, BFA treatment potentiated the effect of BTZ on BNIP3 accumulation (Fig. 3E). Collectively, these results verify that when endolysosomal transport of BNIP3 is perturbed, BNIP3 degradation by the lysosome is severely diminished, although it is re-routed for proteasomal degradation.

## Proteasomal degradation restricts BNIP3 levels but not lysosomal delivery

Based on the data above, we wished to better explore the interplay between proteasomal and lysosomal regulation of BNIP3. Post-translational control of BNIP3 stability was previously reported to depend on the ubiquitin–proteasome system (Thanh Nguyen-Dien et al, 2023; Cao et al, 2023; Alsina et al, 2020; Lobato-Gil et al, 2021; Zheng et al, 2022; Emanuele et al, 2011; He et al, 2022; Poole et al, 2021). Consistent with these reports, our list of genetic effectors recovered numerous UPS factors previously implicated in the regulation of BNIP3, including proteasomal subunits, the NEDD8 conjugation machinery, and the membrane protein extratase valosin-containing protein (VCP). Indeed, we found that BNIP3

protein levels accumulated upon chemical inhibition of either neddylation (MLN-4924) or VCP (CB-5083), particularly under hypoxic conditions (Fig. 4A; S3A,B). More specifically, proteostatic collapse resulted in a striking stabilization of tf-BNIP3 throughout both the mitochondrial and the ER network (Figs. 4B and EV3C). While lysosomal delivery of tf-BNIP3 was generally decreased under these conditions (Fig. 4C), we still noted the presence of RFP-only puncta (Fig. 4B). We interpret this to indicate that proteasomal inhibition dramatically stabilizes tf-BNIP3 protein levels, but it does not arrest lysosomal delivery per se. To test this, we grew parental MDA-MB-231 cells in normoxic or hypoxic conditions, with or without Baf-A1 and/or BTZ for 18 h. We then monitored extracts by immunoblotting (IB) for endogenous levels of BNIP3 and NIX (Fig. 4D). As reported above (Fig. 1D), Baf-A1 significantly stabilized BNIP3 levels regardless of oxygen availability (Fig. 4E). Likewise, BTZ had a generally stabilizing effect that was comparable to or lesser than Baf-A1. When combined, treatment with Baf-A1 and BTZ resulted in the additive accumulation of BNIP3 under both normoxic and hypoxic conditions, further supporting nonoverlapping roles for the proteasome and the endolysosomal system in restricting BNIP3 levels (Fig. 4E).

## BNIP3 dimerization determines its mode of degradation and is required for lysosomal delivery

The soluble portion of BNIP3 (residues 1–163) contains several known motifs and domains. BNIP3 contains a canonical LC3-interacting region (LIR) motif required for mitophagy (Zhu et al, 2013). In addition, it contains a PEST domain, a BH3 domain, and an extended "conserved region" adjacent to the BH3 domain (Fig. 5A) (Yasuda et al, 1998; Chinnadurai et al, 2009). To elucidate the features within BNIP3 that are required for its lysosomal delivery, we performed a structure-function analysis using our tf-reporter system (Fig. 5A). To this end, we transiently expressed tf-BNIP3 variants and monitored red:green ratio as a proxy for lysosomal delivery. As a control for expression levels, we monitored RFP intensity across samples and note that, while transduction efficiency varied, median RFP intensity was comparable between constructs and did not correlate with any reported effect (Fig. EV4A). Mutation of the LIR motif (W18A/L21A) or truncation of the LIR motif (aa30-end) had little effect on the lysosomal delivery of BNIP3. Similarly, a phosphomimetic mutation near the LIR motif, BNIP3$^{S17E}$, that enhances LC3 binding did not increase lysosomal delivery (Fig. 5B) (Zhu et al, 2013). Additional truncations of the PEST domain (aa82-end or aa104-end) also had minimal effect on flux. Subsequent deletion of the BH3 domain (aa 117-end) partially diminished lysosomal delivery although delivery was still active (Fig. 5B). The BH3 domain also has been

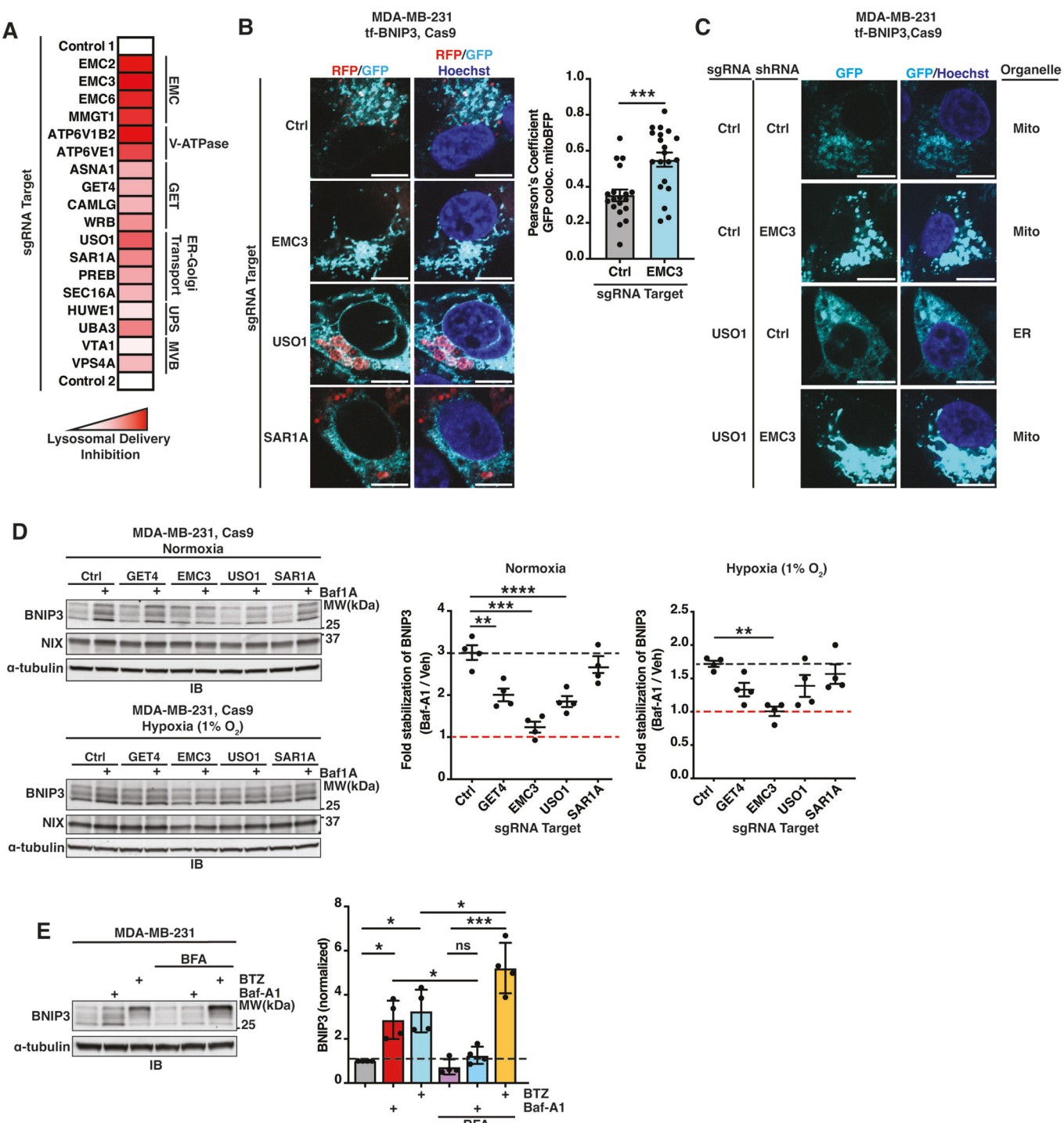

implicated in the proteasomal regulation of the BNIP3, which we confirmed (Fig. EV4B) (Poole et al, 2021). Only a near-complete truncation of the soluble domain (aa137-end), which also eliminates the conserved region, showed dramatic inhibition of lysosomal delivery. Concordant with its decrease in lysosomal delivery, the aa137-end truncation exhibited an increasingly mixed mitochondrial/ER localization pattern including a significant signal on the perinuclear membrane (Fig. EV4C). We conclude that both the conserved region and, to a lesser extent, the BH3 domain,

influence the lysosomal trafficking of BNIP3, likely by facilitating ER export.

As a representative tail-anchored protein, BNIP3 also possesses a single, C-terminal, transmembrane domain that is essential for its localization, insertion into membranes, and dimerization (Vande Velde et al, 2000; Chen et al, 1997; Yasuda et al, 1998). As dimerization has been routinely tied to the functionality of both BNIP3 and NIX (Bocharov et al, 2007; Marinković et al, 2020; Ray et al, 2000), we generated two transmembrane mutants in BNIP3 to

**Figure 3. BNIP3 lysosomal delivery is governed by ER insertion and the secretory pathway.**

(A) MDA-MB-231 cells expressing tf-BNIP3 and Cas9 were transduced with sgRNAs for the indicated genes. The median red:green ratio of each population was used to generate a heatmap. Darker shades of red indicate greater inhibition, with a red:green ratio of 1 taken as the theoretical maximum inhibition. Genes were clustered by related functions. For underlying data, see Fig. EV2A. (B) Representative confocal micrographs of tf-BNIP3-expressing cells transduced with sgRNAs targeting the indicated genes. Pearson's correlation coefficient between GFP and mitoBFP (reflective of mitochondrial localization) was calculated using Coloc2. Bar graphs represent mean $+/-$ SEM. Each data point represents a single cell. Statistical analysis was performed using an unpaired $t$ test. Scale bar: 10 μm; $n = 15$ cells; ***$P<0.001$. (C) Representative confocal micrographs of tf-BNIP3-expressing cells transduced with indicated sgRNA and shRNA. Scale bar: 10 μm. (D) Immunoblotting (IB) of MDA-MB-231-derived extracts from cells transduced with indicated sgRNAs in both normoxia and hypoxia. Where indicated, cells were treated with 100 nM Baf-A1 for 18 h. Shown are representative images from one biological replicate. Quantification of BNIP3 protein stabilization by Baf-A1 treatment was calculated as: (BNIP3$^{Baf-A1}$/tubulin$^{Baf-A1}$)/ (BNIP3$^{DMSO}$/tubulin$^{DMSO}$). Graphs represent the mean $+/-$ SEM from four independent experiments. Black dashed line indicates fold-stabilization of BNIP3 upon Baf-A1 treatment in control cells. Red dashed line demarcates no stabilization. Statistical analysis was performed a one-way ANOVA with Tukey's test. ****$P<0.0001$; ***$P< 0.001$; **$P<0.01$. (E) IB of MDA-MB-231-derived extracts from cells treated with Brefeldin-A (BFA) (1 μM), Baf-A1 (100 nM), or bortezomib (BTZ) (100 nM) for 18 h. Shown are representative images from one biological replicate. Bar graphs represent mean $+/-$ SEM from four independent experiments. All protein levels were normalized to α-tubulin. Statistical analysis was performed using a one-sample $t$ test to the normalized control and an unpaired Student's $t$ test between experimental samples, Veh (DMSO) ***$P<0.001$; *$P< 0.05$; ns not significant. Source data are available online for this figure.

investigate the role of dimerization in lysosomal delivery. First, we generated a frequently used serine-to-alanine mutation (S172A), which disrupts intermonomer side chain hydrogen bonding (Sulistijo and MacKenzie, 2009; Lawrie et al, 2010; Sulistijo et al, 2003). Second, we swapped the positions of leucine-179 and glycine-180 (LG swap). These two residues are part of the transmembrane GxxxG motif required for dimer formation (Lawrie et al, 2010; Sulistijo et al, 2003; Yasuda et al, 1998). The LG swap mutation disrupts the motif registrar while maintaining the overall hydrophobicity of the TMD segment. When expressed in HEK293T cells, both mutations disrupted the formation of SDS-resistant dimers (Fig. 5C). In a corresponding functional assay, both dimer mutations disrupted BNIP3 delivery to lysosomes (Fig. 5D). We then monitored the cellular localization of the two dimerization mutants to see where they arrested. Surprisingly, our dimerization mutants were differentially localized. BNIP3$^{S172A}$ localized in a reticular, ER-like pattern (Fig. 5E). We anticipate this shift in localization is due to the changing hydrophobicity of the transmembrane domain, as hydrophobicity is a primary determinant for tail-anchored protein targeting (Wattenberg et al, 2007). In contrast, the LG swap of the GxxxG motif, which does not affect hydrophobicity, remained primarily on mitochondria (Fig. 5E).

Failure of dimerization mutants to traffic to the lysosome suggests that dimerization is an important aspect of BNIP3 trafficking. However, localization discrepancies limited our ability to cleanly interpret these results. To solidify the role of dimerization in BNIP3 trafficking, we turned to an inducible dimerization system (Fig. 5F) (Clackson et al, 1998). Since BNIP3 dimerization is mediated through its transmembrane segment, we wished to fuse a DmrB artificial dimerization domain as close to the transmembrane lesion as possible. Given the topology of TA proteins, C-terminal tagging is not possible. Therefore, we tagged the N-terminus of the shortest functional BNIP3 truncation (BNIP3$^{117-end}$). Next, we transiently expressed tf-BNIP3$^{117-end}$ or the dimerization variants (S172A and LG swap) with or without an in-frame DmrB dimerization domain. We then incubated cells with AP20187, a small molecule that induces homodimerization of the DmrB domain, and monitored red:green ratio as a proxy for flux. The homodimerizer molecule did not affect the red:green ratio for wild-type BNIP3 (tf-DmrB-BNIP3$^{117-end}$) or any constructs lacking the DmrB domain (Figs. 5G and EV4D). However, ectopic dimerization dramatically rescued the red:green ratio of the DmrB-fused S172A mutant (Fig. 5G). Importantly, Baf-A1 attenuated

this increase, confirming the increase was due to lysosomal delivery. In contrast, the mitochondrially restricted LG swap mutant was minimally responsive to the homodimerizer (Fig. 5G). Collectively, these data illustrate that dimerization within the ER membrane is a required aspect of BNIP3 trafficking to the lysosome.

Within this model, what is the fate of unassembled BNIP3 monomers? To address this, we employed the global protein stability (GPS) cassette, a reporter used to study proteasomal degradation and protein degrons (Koren et al, 2018). In brief, the cassette contains an RFP fluorophore, followed by an internal ribosome entry site (IRES) and a GFP fluorophore tethered to BNIP3 (Fig. 5H). This results in the expression of two polypeptides: a cytosolic RFP and a GFP-BNIP3 fusion protein. The relative stability of individual GFP-BNIP3 variants can then be assessed by red:green ratio. This approach is methodologically similar to the tf-BNIP3 reporter. However, the output better incorporates the effects of proteasomal regulation. By this approach, we observe that dimerization mutants are destabilized compared to wild type (Fig. 5I). Consistent with our tf-BNIP3 reporter, treatment with Baf-A1 stabilized wild-type BNIP3 but did not stabilize either dimer mutant. However, both dimer mutants were dramatically stabilized by a chemical inhibitor of the proteasome, BTZ, indicating they are selectively targeted by the UPS. Similarly, chemical inhibition of the ubiquitin-activating enzyme, UBA1, with TAK-243 also reduced protein turnover, suggesting ubiquitin-mediated degradation. Chemical inhibition of VCP (CB-5083) selectively stabilized ER-localized monomers, highlighting that proteostatic regulation of BNIP3 is differentially governed at the ER and mitochondrial membranes (Fig. 5I). Collectively, these results suggest that BNIP3 dimerization state and organelle localization determines the mode of degradation.

## Lysosomal delivery of BNIP3 is distinct from BNIP3-mediated mitophagy

Autophagy receptors are frequently degraded in tandem with their cargo. The observation that BNIP3 flux is largely autophagy-independent opposes this paradigm, leading us to more specifically evaluate the relationship between BNIP3 flux and BNIP3-mediated mitophagy. To distinguish the lysosomal delivery of BNIP3 from BNIP3-mediated mitophagy, we turned to an established mitophagy reporter, mt-Keima (Vargas et al, 2019). This reporter

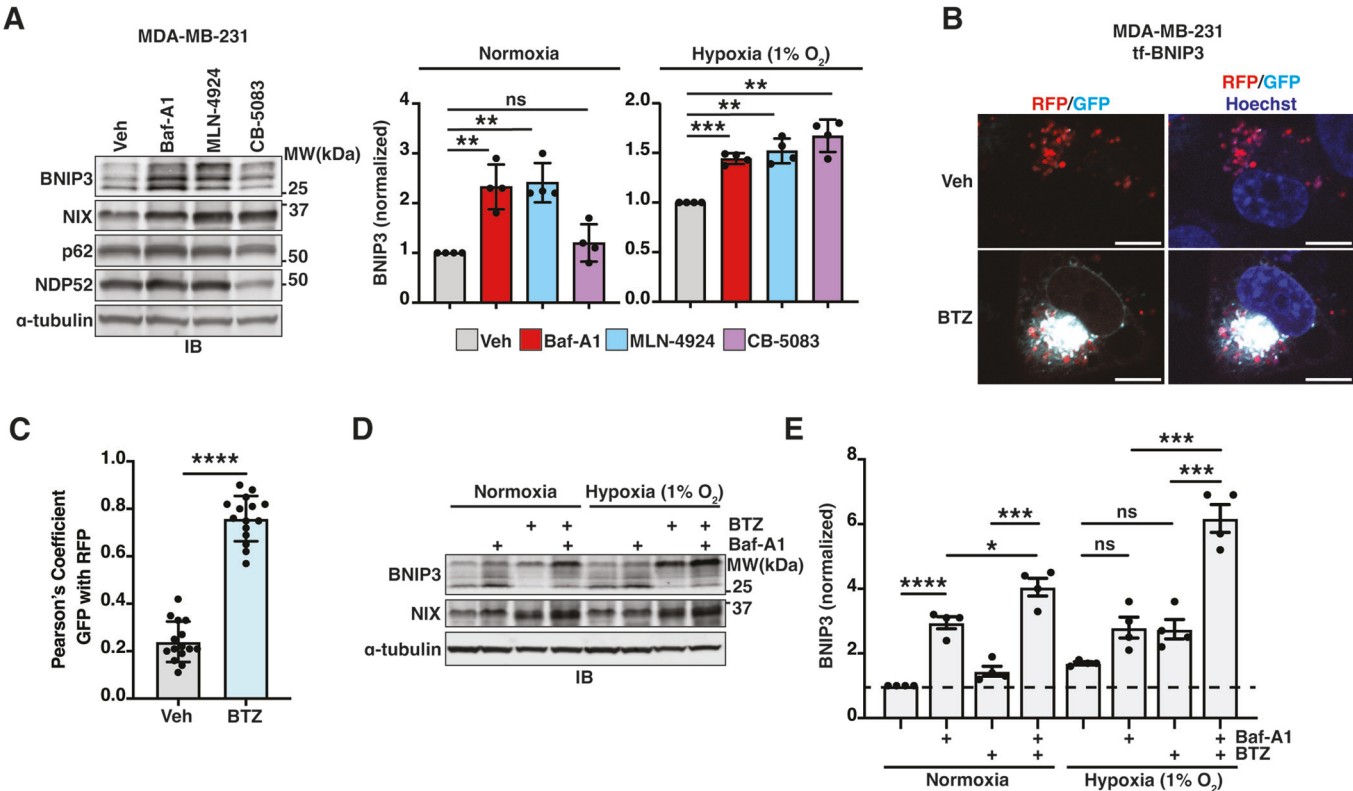

**Figure 4. The proteasome is required for efficient BNIP3 degradation, but not lysosomal delivery.**

(A) Immunoblotting (IB) of MDA-MB-231-derived extracts from cells treated with vehicle (DMSO), Baf-A1 (100 nM), MLN-4924 (1 μM), and CB-5083 (1 μM) for 18 h. Shown are representative images from one biological replicate (for hypoxia, see Fig. EV3A). Bar graphs represent mean +/− SEM from four independent experiments. All protein levels were normalized to α-tubulin. Statistical analysis was performed using a one-sample *t* test to the normalized control ***P<0.001, **P< 0.01; ns is not significant. (B) Representative confocal micrographs of fixed tf-BNIP3-expressing cells treated with vehicle (DMSO) or bortezomib (BTZ) (100 nM) for 18 h. Scale bar: 10 μm. (C) Pearson's correlation coefficient between GFP and RFP was calculated using Coloc2. Bar graph represents mean +/− SEM. Each data point represents a single cell. Statistical analysis was performed using an unpaired Student's *t* test. Scale bar: 10 μm; *n* = 15 cells; ****P<0.0001. (D) Immunoblotting (IB) of MDA-MB-231-derived extracts from cells grown in normoxia and hypoxia and treated with DMSO, Baf-A1 (100 nM), and/or bortezomib (BTZ) (100 nM) for 18 h. Shown are representative images from one biological replicate. (E) Quantification of BNIP3 protein accumulation from (D). Bar graphs represent mean +/− SEM from four independent experiments. All protein levels were normalized to α-tubulin. Statistical analysis was performed using a one-way ANOVA with Dunnett test and an unpaired Student's *t* test between experimental samples. ****P<0.0001; ***P<0.001; *P<0.05; ns not significant. Source data are available online for this figure.

encodes a cytochrome c oxidase signal sequence fused to a pH-sensitive fluorescent protein, mKeima. In MDA-MB-231 cells, we observed moderate basal flux of mt-Keima (15%), as normalized to Baf-A1 treatment (Fig. 6A). Knockout of *ATG9A* did not inhibit lysosomal delivery of the mt-Keima reporter (Fig. EV5A, compare sgCtl vs sgATG9A). Thus, basal flux in MDA-MB-231 cells is largely independent of autophagy. Autophagy-independent delivery of mitochondrial content to lysosomes is likely due to mitochondrial-derived vesicles (König et al, 2021; Towers et al, 2021; Neuspiel et al, 2008). Therefore, we dubbed the readout for the mt-Keima reporter as "mitoflux" to incorporate autophagic and non-autophagic turnover of mitochondria.

To assess how BNIP3 variants influence mitophagy, we took advantage of the fact that BNIP3 overexpression induces mitophagy (Kim et al, 2021; Quinsay et al, 2010; Lee et al, 2011). We transiently expressed BFP-BNIP3 variants or a BFP empty vector control in MDA-MB-231 cells expressing mt-Keima. As a control for expression levels, we monitored BFP intensity across samples and note that, while transduction efficiency varied, median BFP

intensity was comparable between constructs and did not correlate with any reported effect (Fig. EV5B). Expression of wild-type BNIP3 notably increased the percentage of mitoflux+ cells as compared to the BFP-only control (31.7% vs 14.7%) (Fig. 6A). Combination treatment with hypoxia led to an additive induction of mitoflux (Fig. EV5C). We then tested mitophagy induction by BNIP3 mutants. Consistent with previous reports (Poole et al, 2021; Zhu et al, 2013), the phosphomimetic S17E mutant modestly increased mitoflux above wild-type BNIP3 (mean values 18.3% empty vs 34.7% WT vs 42% S17E, *P*<0.0001) (Fig. 6B). Correspondingly, the double LIR mutant (W18/L21A) failed to enhance mitoflux as did all tested truncations of BNIP3 (Fig. 6B). We note that these data strongly contrast with the trends observed for endolysosomal trafficking of BNIP3 (compare Figs. 6B and 5B), confirming that lysosomal delivery and mitophagy are functionally separable features of BNIP3.

In addition to the LIR motif, the transmembrane helix has been intermittently implicated in BNIP3- and NIX-mediated mitophagy (Marinković et al, 2020; Hanna et al, 2012). We found that the

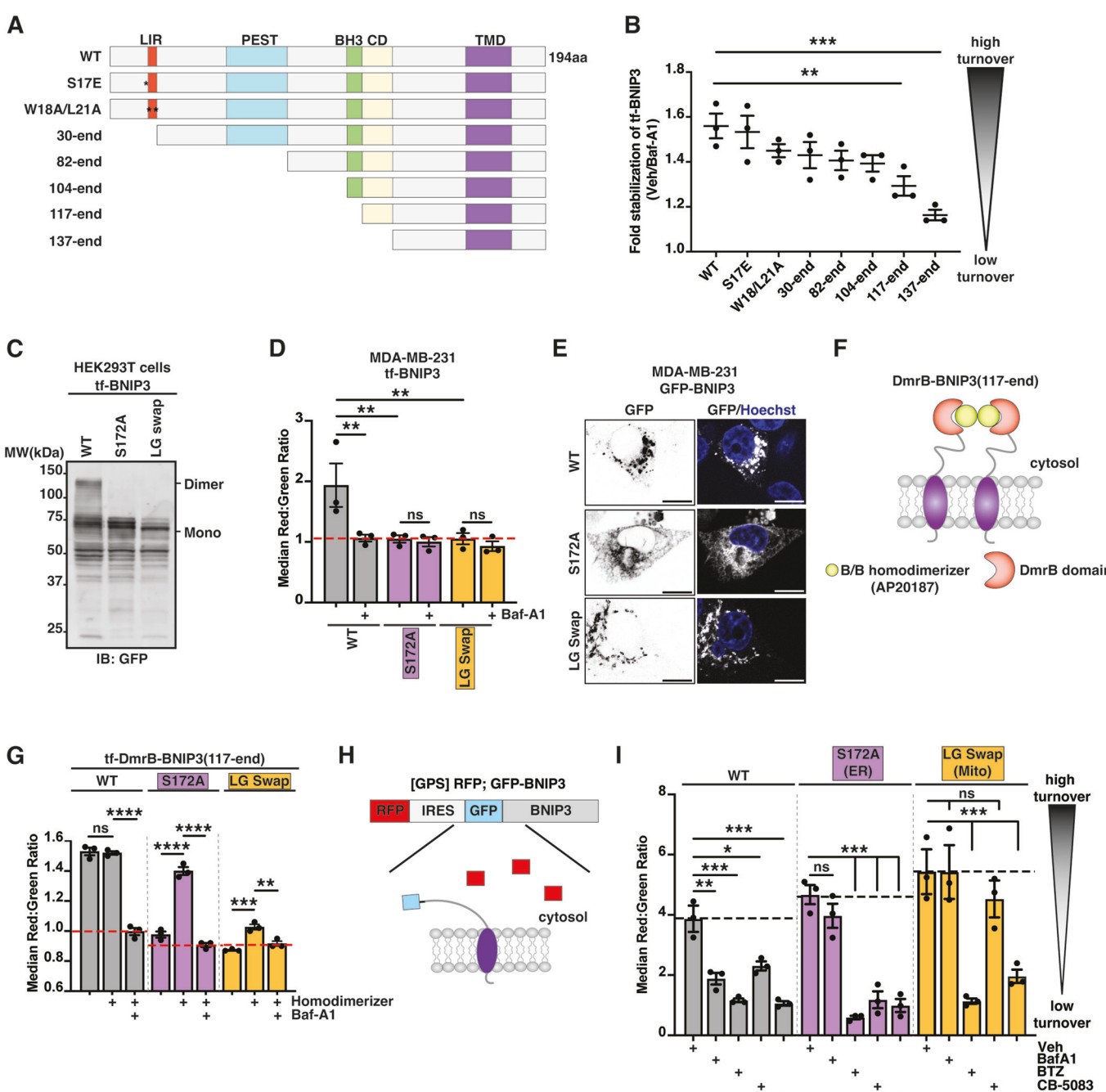

mitochondrially localized LG swap mutant induced mitoflux comparably to wild type while the ER-localized S172A mutant did not (Fig. 6B). This discrepancy suggests that BNIP3-induced mitophagy is not strictly dependent on dimerization of the BNIP3 TMD. To test this, we swapped the endogenous BNIP3 transmembrane domain with the transmembrane domain from an unrelated, monomeric, mitochondrial TA protein, Fis1 (hereafter BNIP3(FIS1$^{TMD}$)). A tf-BNIP3(FIS1$^{TMD}$) chimera failed to traffic to the lysosome, presumably due to abolished dimerization and/or diminished ER localization (Fig. 6C). However, BNIP3(FIS1$^{TMD}$) induced mitophagy comparable to wild-type BNIP3 (Figs. 6D and EV5D) (Hanna et al, 2012). We cannot dismiss that overexpression of BNIP3(FIS1$^{TMD}$) may bypass the need for BNIP3

dimerization in mitophagy induction. However, these data indicate that the cytosolic portion of the BNIP3 protein tethered to the OMM is sufficient to induce mitophagy and the native BNIP3 TMD domain is not required.

Are the aforementioned, BNIP3-dependent, changes in mitoflux sufficient to affect cellular physiology? To evaluate the functional consequences of this mitophagy, we analyzed metabolic flux in cells expressing BNIP3 or its variants. Ectopic expression of BNIP3 truncations decreased oxygen consumption rates (OCR) and increased extracellular acidification rate (ECAR) commensurate with their ability to induce mitophagy (Fig. 6E,F). Thus, the levels of mitophagy induced by ectopic BNIP3 expression are sufficient to

**Figure 5. BNIP3 dimerization determines mode of degradation and is required for lysosomal delivery.**

(A) Domain organization of BNIP3 (NP_004043.4) and derived variants. LC3 LC3-interacting region, PEST PEST domain, BH3 BH3 domain, CR conserved region, TMD transmembrane domain. Asterisks in the schematic indicate individual amino acid residue positions. (B) Dot plot representing fold-stabilization of tf-BNIP3 variants in response to Baf-A1. Stabilization was calculated as a ratio of median red:green ratios (DMSO/Baf-A1). A ratio of 1 represents no lysosomal delivery. Plots represent the mean $+/-$ SEM from three independent experiments. Statistical analysis was performed using a one-way ANOVA with Dunnett test. ***$P$<0.001; **$P$<0.01. (C) Immunoblotting (IB) of HEK293T-derived extracts transiently expressing the indicated tf-BNIP3 variants. Monomeric and dimeric species are indicated. (D) MDA-MB-231 cells were transduced with the indicated tf-BNIP3 variants. Red:green ratio was analyzed by flow cytometry 48 h post transduction. Cells were incubated with vehicle (DMSO) or Baf-A1 (100 nM) for 18 h before analysis. The red dotted line across all samples corresponds to red:green ratio of wild-type (WT) cells inhibited with Baf-A1 ($n$ >10,000 cells). Bar graphs represent mean $+/-$ SEM from three independent experiments. Statistical analysis was performed using a two-way ANOVA with Dunnett's test. **$P$<0.01; ns not significant. (E) Representative confocal micrographs of MDA-MB-231 cells transduced with indicated GFP-BNIP3 variants. Scale bar, 10 μm. (F) Schematic of the DmrB-based inducible dimerization system using the 117-end variant of BNIP3. (G) MDA-MB-231 cells were transduced with the indicated tf-BNIP3[117-end] variants. Red:green ratio was analyzed by flow cytometry 48 h post transduction. Cells were treated with Baf-A1 (100 nM) and/or B/B homodimerizer (0.5 μM) for 6 h prior to performing flow cytometry. Bar graphs represent mean $+/-$ SEM from three independent experiments. The red dotted line across all samples corresponds to cells inhibited with Baf-A1 ($n$ >10,000 cells). Statistical analysis was performed using a two-way ANOVA with Dunnett' test. ****$P$<0.0001; ***$P$<0.001; **$P$< 0.01; ns not significant. (H) Schematic of global protein stability (GPS) cassette fused to BNIP3. IRES, internal ribosome entry site. (I) MDA-MB-231 cells were transduced with the indicated [GPS] RFP; GFP-BNIP3 variants. Red:green ratio was analyzed by flow cytometry 48 h post transduction. Cells were treated with vehicle (DMSO), Baf-A1 (100 nM), BTZ (100 nM), CB-5083 (1 μM), TAK-243 (an inhibitor of the ubiquitin-activating enzyme [UAE], 1 μM) for 18 h prior to performing flow cytometry. Bar graphs represent mean $+/-$ SEM from three independent experiments. The black dotted line across each sample group corresponds to the basal red:green ratio of mock-treated cells ($n$ >10,000 cells). Statistical analysis was performed using a two-way ANOVA with Dunnett' test. ****$P$< 0.0001; ***$P$< 0.001; **$P$<0.01; *$P$<0.05; ns not significant. Source data are available online for this figure.

drive changes in global energy metabolism. Similarly, the constitutively mito-localized BNIP3(FIS1[TMD]) variant decreased OCR and increased ECAR comparable to wild type (Fig. EV5E), consistent with the thesis that the conserved TMD of BNIP3 is not strictly required for the mitophagic activity of BNIP3 but correlates instead with the cell's ability to employ the endolysosomal system to regulate BNIP3 localization and function.

## The endolysosomal and proteasomal systems confine BNIP3 levels to suppress basal mitophagy

While BNIP3-induced mitophagy affects cellular physiology, these results were obtained through ectopic expression of BNIP3. What contribution does endogenous BNIP3 make towards mitoflux and cellular metabolism, and how does regulation by the UPS and the endolysosomal system impinge upon this system? To begin, we used mt-Keima MDA-MB-231 cells to validate the contribution of endogenous BNIP3 to mitophagy in response to well-documented queue, hypoxia. Indeed, we found that hypoxia-responsive increases in mitoflux were dependent on endogenous *ATG9A* and *BNIP3* (Appendix Fig. S1A). Using this reporter system, we then induced broad proteostatic collapse in mt-Keima cells using bortezomib, CB-5083, or MLN-4924. Mitoflux increased upon treatment with all three inhibitors (Fig. 7A). Critically, the induction of mitoflux by proteostatic collapse was dependent on *ATG9A* and *BNIP3*, consistent with a mitophagy-specific defect (Fig. 7B; Appendix Fig. S1B). Moreover, deletion of the mitochondrial insertases MTCH1 and MTCH2 also attenuated BTZ-induced mitophagy (Appendix Fig. S1C), reflective of their role in establishing BNIP3 in the OMM (Appendix Fig. S1D). Proteostatic collapse similarly induced BNIP3-dependent mitophagy in an alternative cell line, U2OS, albeit (1) U2OS cells displayed a slightly greater basal mitophagy and (2) mitophagy in these cells also was also dependent on NIX, consistent with previous findings (Appendix Fig. S1D) (Gok et al, 2023; Cao et al, 2023; Thanh Nguyen-Dien et al, 2023).

We then interrogated the role of the endolysosomal system in regulating mitoflux. To this end, we transduced Cas9-expressing mt-Keima cells with an sgRNA targeting *EMC3*. Knockout of

*EMC3* induced mitoflux relative to a control sgRNA (~17% vs ~34%, $P$<0.05) and combining *EMC3* deletion with proteasome inhibition had additive effects on mitoflux (30% vs 47.8%, $P$<0.001) (Fig. 7C; Appendix Fig. S1E). Critically, while EMC3 deletion elevated mitoflux, this effect was strongly dependent on BNIP3, as concurrent knockdown of BNIP3 with a short hairpin RNA (shBNIP3) returned mitoflux values to baseline (Fig. 7D). Incubation of *EMC3* knockout cells in hypoxia also had additive effects on mitophagy stimulation (Appendix Fig. S1F) suggesting that ER insertion of BNIP3 regulates mitophagy across a range of mitophagy inducing conditions. In sum, these data are consistent with the UPS and the endolysosomal system making nonoverlapping contributions towards restricting endogenous BNIP3 mitoflux, and they establish BNIP3 as a node of integration for endolysosomal and proteasomal regulation of mitophagy (Fig. 7E).

# Discussion

Immense efforts have been directed toward understanding PINK1/Parkin-mediated mitophagy (Nguyen et al, 2016). However, less is known about other mitophagy processes, including BNIP3- and NIX-mediated mitophagy. Early models for BNIP3-mediated mitophagy centered on its transcriptional control, particularly in response to hypoxic stress (Bellot et al, 2009; Zhang et al, 2008). Recently, these models have been appended to account for posttranslational control by the ubiquitin–proteasome system (Zheng et al, 2022; Alsina et al, 2020; Thanh Nguyen-Dien et al, 2023; Cao et al, 2023; He et al, 2022; Poole et al, 2021; Wu et al, 2021). Using an unbiased, genome-wide CRISPR screen, we similarly identified a role for the ubiquitin–proteasome system in regulating BNIP3, providing independent support for these models. However, prior reports do not fully account for BNIP3 dynamics in the cell.

As is documented for many autophagy receptors, lysosomal degradation of BNIP3 was presumed to be primarily through autophagy. Here, we demonstrate an alternative mode of BNIP3 degradation that is lysosome-mediated yet autophagy-independent. This alternative lysosomal delivery accounts for the vast majority of BNIP3's lysosome-mediated turnover, even upon mitophagy

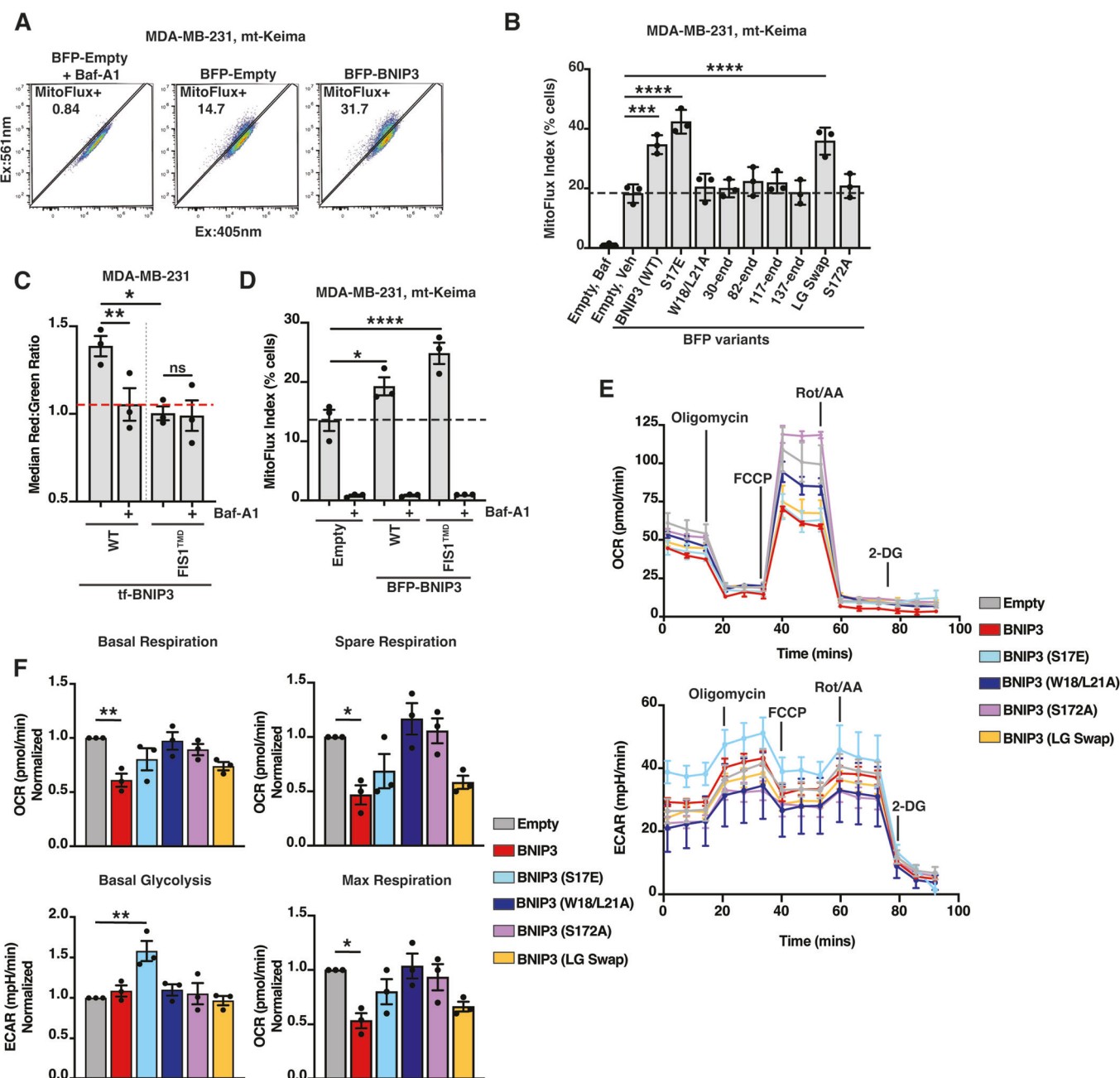

induction. Degradation of BNIP3 by mitophagy likely accounts for a portion of BNIP3 turnover in the cell, but it is below the level of detection in our system. Several mitochondrial proteins have been reported to escape from mitochondria during mitophagy by relocalizing to the ER (Saita et al, 2013; Bhujabal et al, 2017). Whether BNIP3 avoids autophagic degradation by a similar mechanism remains an intriguing possibility. In any case, lysosomal delivery of BNIP3 and/or NIX is an unexpectedly poor correlate for BNIP3/NIX-mediated mitophagy.

Our data indicate that the endolysosomal system functions independently of proteasomal regulation to further modulate levels and localization of BNIP3. When both mechanisms are disrupted, we see an additive increase in BNIP3 protein levels with a

corresponding increase in mitophagy (Figs. 4B and 6B). Inhibition of ER insertion does not result in the overall accumulation of BNIP3 due to the compensatory effects of the proteasome. Regardless, the deletion of EMC components spatially restricts BNIP3 to the mitochondria, elevating mitophagy. In short, while BNIP3 can be cleared by parallel and partially compensatory quality control pathways, non-autophagic lysosomal degradation of BNIP3 is a strong posttranslational modifier of BNIP3 function in both normoxic and hypoxic conditions.

With a new perspective on BNIP3 regulation, we took a structure-function approach to clarify the role of multiple conserved regions of BNIP3 including the LIR, the BH3 domain and its adjacent "conserved region", and the C-terminal TMD. The N-terminal LIR

**Figure 6. Lysosomal delivery is distinct from BNIP3-mediated mitophagy.**

(A) MDA-MB-231 cells expressing mt-Keima were transduced with BFP-empty or BFP-BNIP3 and analyzed by flow cytometry 48 h post transduction. Cells were incubated with vehicle (DMSO) or Baf-A1 (100 nM) for 18 h before analysis. Baf-A1 treatment was used to define "MitoFlux+", indicative of cells turning over mitochondria. *n* >10,000 cells. (B) MDA-MB-231 cells expressing mt-Keima were transduced with indicated BFP-BNIP3 variants and BFP-positive cells were analyzed for Mitoflux as in (A). Bar graphs represent mean +/− SEM from three independent experiments. Statistical analysis was performed using a one-way ANOVA with Dunnett's test. ****$P<0.0001$; ***$P<0.001$. (C) MDA-MB-231 cells were transduced with wild-type (WT) tf-BNIP3 or tf-BNIP3(FIS1$^{TMD}$) and analyzed by flow cytometry 48 h post transduction. Cells were incubated with vehicle (DMSO) or Baf-A1 (100nM) for 6 h before analysis. Bar graphs represent mean +/− SEM from three independent experiments. The red dotted line across all samples corresponds to red:green ratio of wild-type cells inhibited with Baf-A1 (*n* >10,000 cells). Statistical analysis was performed using a one-way ANOVA with Dunnett's test. **$P<0.01$; *$P<0.05$; ns not significant. (D) MDA-MB-231 cells expressing mt-Keima were transduced with indicated BFP-BNIP3 variants and analyzed for Mitoflux as in (A). Bar graphs represent mean +/− SEM from three independent experiments. The black dotted line across each sample group corresponds to the basal red:green ratio of mock-treated cells (*n* >10,000 cells). Statistical analysis was performed using a one-way ANOVA with Dunnett's test. ****$P<0.0001$; *$P<0.05$. (E) Representative Seahorse Flux Analyzer graph of MDA-MB-231 cells that were transduced with indicated the BFP-BNIP3 variants and analyzed for oxygen consumption rate (OCR) and extracellular acidification rate (ECAR) 48 h post transduction. Graphs represent the mean +/− SEM from five technical replicates. Values were normalized by BCA protein assay. (F) Quantification of basal respiration, basal glycolysis, spare respiration, and max respiration from (E). Bar graphs represent mean +/− SEM from three independent experiments. Statistical analysis was performed using a one-way ANOVA with Dunnett's test. **$P<0.01$; *$P<0.05$. Source data are available online for this figure.

motif (ϕ-x-x-ψ) is a phospho-regulated motif governing the interaction of BNIP3 with ATG8-family proteins (Zhu et al, 2013; Hanna et al, 2012). As previously reported, we find that mutation of the LIR motif fully ablates BNIP3-mediated mitophagy. However, this region has no bearing on the lysosomal delivery of BNIP3, reinforcing BNIP3's autophagy-independent flux. In contrast, BNIP3's atypical BH3 domain has a modest effect on lysosomal delivery. Unlike canonical BH3 domains, this domain does not appear to function in cell death (Kubli et al, 2007; Landes et al, 2010; Ray et al, 2000; Zhang et al, 2009). Rather, it was recently implicated in the proteasome-mediated stability of BNIP3 (Poole et al, 2021). Our data support this interpretation, as truncation through the BH3 domain rendered BNIP3 resistant to proteasome inhibition by bortezomib (Fig. EV4A). Continuous with the BH3 domain is an 11 amino acid conserved region of unknown function (Ray et al, 2000). We find that truncation through this conserved region strongly disrupts the endolysosomal trafficking of BNIP3, leading to a mixed ER/mitochondria distribution (Fig. EV4C). While we cannot exclude other functions for this region, we speculate that its conservation is a function of its role in the endolysosomal regulation of BNIP3. Finally, the C-terminal TMD of BNIP3 contains a GxxxG motif required for homodimerization (Lawrie et al, 2010; Sulistijo and MacKenzie, 2006). Disruption of this motif ablated the formation of SDS-resistant dimers but did not affect mitophagy, as measured by a highly quantitative mt-Keima assay. Similarly, overexpression of a chimera protein, BNIP3(Fis1$^{TMD}$), induced mitophagy comparable to wild-type BNIP3, although BNIP3(Fis1$^{TMD}$) was no longer subject to endolysosomal degradation. On the surface, these data contrast with previous models, wherein oligomerization governs the activation of autophagy receptors. However, we cannot make definitive claims about the role of dimerization in mitophagy due to several possible factors. First, overexpression of BNIP3 might bypass the need for BNIP3 dimerization in mitophagy induction. Second, the soluble domain of BNIP3 could retain sufficient self-association for mitophagy (Kubli et al, 2008). Third, the clustering of BNIP3 might be driven through interaction with a soluble autophagy scaffold (Margolis et al, 2020). Interestingly, the glutathione peroxidase GPX4, gatekeeper of ferroptosis and oxidative membrane stress, was found as a strong effector in our BNIP3 flux screen, consistent with a role for oxidative stress in regulating BNIP3 dimerization in vivo (Kubli et al, 2008; Gao et al, 2019). Additional studies will clearly be needed to fully disentangle the regulation and role of BNIP3 dimerization. Yet, our data clearly indicate that the TMD of BNIP3 is dispensable for

BNIP3-mediated mitophagy. Going forward, the ability to functionally separate the mitophagy and ER-trafficking activities of BNIP3 provides a foundation for future testing of more specific hypotheses regarding BNIP3 function in vivo.

More broadly, our findings have general implications for membrane protein quality control. Organelle identity and function are largely defined by the unique composition of each organelle's constituent proteins. At first glance, the dynamic exchange of membrane proteins between organelles would appear paradoxical. However, growing evidence suggests that kinetically driven cycles of insertion and extraction—rather than a single, high-fidelity insertion event—best explain the observed, steady-state partitioning of many membrane proteins (Hansen et al, 2018; McKenna et al, 2020; Xiao et al, 2021; Matsumoto and Endo, 2022; Matsumoto et al, 2019; McKenna et al, 2022). Perturbing this cycle results in the aberrant accumulation of TA proteins at incorrect membranes. Extending these observations, we find constitutive delivery of BNIP3, a model TA protein, to the ER in the absence of any genetic perturbation. In this context, BNIP3 delivery is strongly dependent on the EMC, with the GET complex playing a lesser role. This is congruent with the observation that mitochondrial TA proteins and EMC substrates possess similarly hydrophilic TMDs, although previous studies suggest the GET pathway can intercede when confronted with a significant buildup of non-optimal TA substates (Xiao et al, 2021; Vitali et al, 2018).

Mitochondrial TA proteins that mislocalize to the ER are recognized by ATP13A1 (Spf1 in yeast), an ER-resident ATPase functionally analogous to ATAD1/Msp1 in the OMM (McKenna et al, 2022, 2020). Supporting its role as a TA protein extractase, deletion of ATP13A1/Spf1 results in the accumulation of mitochondrial TA proteins on the ER (Qin et al, 2020; Krumpe et al, 2012; Dederer et al, 2019; McKenna et al, 2020). Why, then, might cells require an alternative ER clearance system such as the endolysosomal pathway employed for BNIP3? An emerging paradigm for TA protein extractases is that orphan TA proteins are preferred substrates (Weir et al, 2017; Dederer et al, 2019). This includes excess or mislocalized TA proteins that fail to incorporate into stable, higher-ordered complexes. Because BNIP3 self-associates, we anticipate that the formation of a stable homodimer renders BNIP3 resistant to ATP13A1-mediated extraction and necessitates an alternative quality control mechanism. At the same time, BNIP3 dimerization is strictly required for lysosomal delivery.

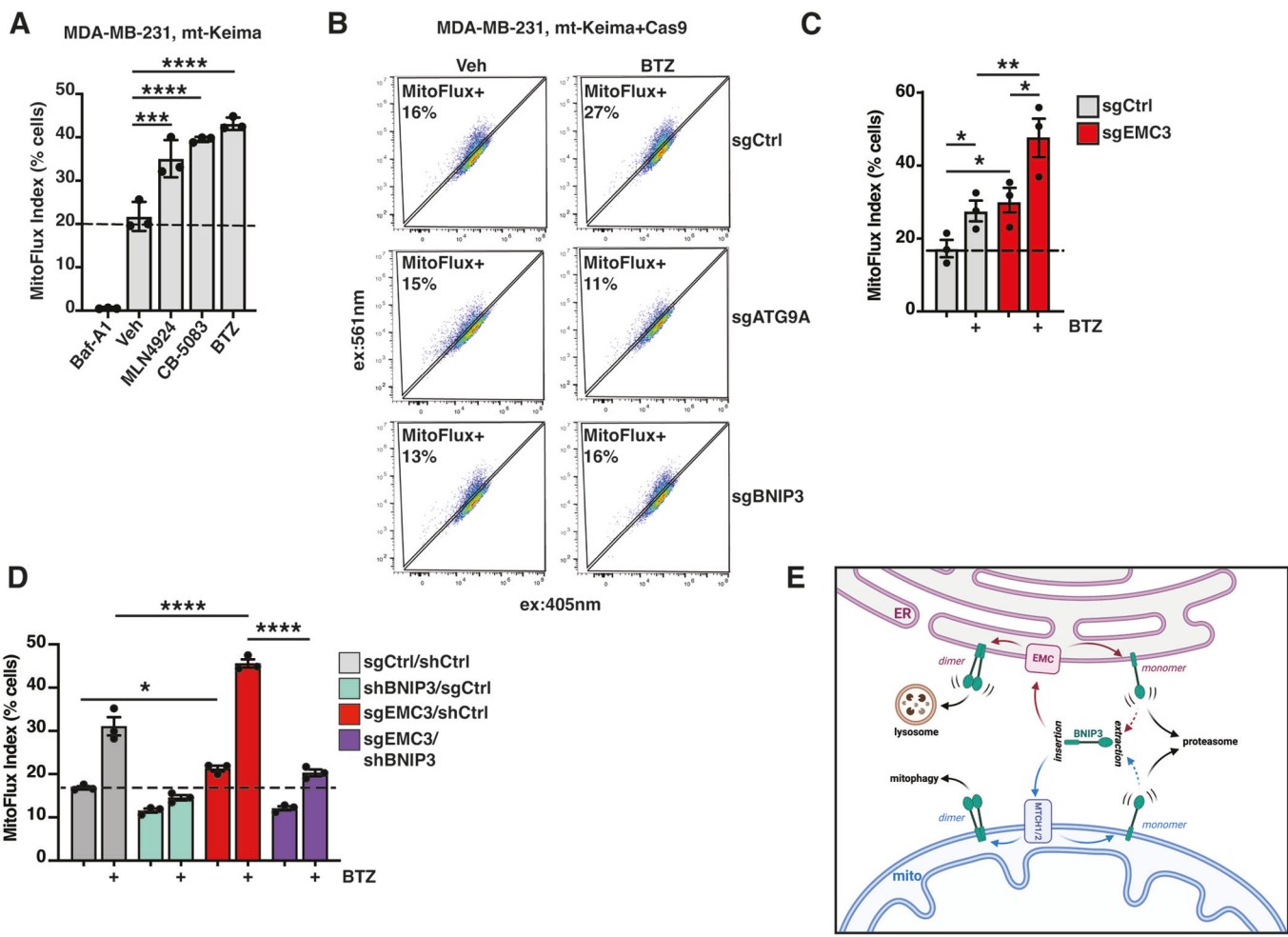

**Figure 7. Endolysosomal and proteasomal systems confine BNIP3 levels to suppress basal mitophagy.**

(A) MDA-MB-231 mt-Keima cells treated with vehicle (DMSO), Baf-A1 (100 nM), MLN-4924 (1 µM), and CB-5083 (1 µM) for 24 h prior to analysis by flow cytometry. Bar graphs represent mean +/− SEM from three independent experiments. Statistical analysis was performed using a one-way ANOVA with Dunnett's test. ****$P<0.0001$; ***$P<0.001$. (B) MDA-MB-231 cells expressing mt-Keima were transduced the indicated sgRNAs. Cells were incubated with vehicle (DMSO) or Baf-A1 (100 nM) for 18 h prior to analysis by flow cytometry. ($n$ >10,000 cells). (C) MDA-MB-231 cells expressing mt-Keima were transduced the indicated sgRNAs. Cells were incubated with vehicle (DMSO) or BTZ (100nM) for 18 h prior to flow cytometry. Bar graphs represent mean +/− SEM from three independent experiments. Statistical analysis was performed using two-way ANOVA with Tukey's post test. **$P<0.01$; *$P<0.05$. (D) MDA-MB-231 cells expressing mt-Keima were transduced the indicated sgRNAs and shRNAs. Cells were incubated with vehicle (DMSO) or BTZ (100 nM) for 18 h prior to flow cytometry. Bar graphs represent mean +/− SEM from three independent experiments. Statistical analysis was performed using two-way ANOVA with Tukey's post test. ****$P<0.0001$; *$P<0.05$. (E) Presumptive model for endolysosomal regulation of mitophagy. Kinetic proofreading enforces the ultimate localization profile of BNIP3 despite limited targeting information, with the lysosome and proteasome serving as sinks that regulate available BNIP3. Solid lines represent processes reported in this study. Dashed lines reflect the presumed cycling of TA proteins. Model generated in BioRender (Adapted from McKenna et al, 2020). Source data are available online for this figure.

Therefore, we propose that self-association enforces a switch between proteasomal and lysosomal degradation routes. In further support of this model, we found accelerated clearance of BNIP3 monomers at both mitochondria and the ER, in a strictly proteasome-dependent manner (Fig. 5I). The role of ATAD1 and ATP13A1 in destabilizing and/or shuttling these BNIP3 monomers is beyond the scope of this work but will be an important area of future study. However, we find that misinsertion into the ER membrane (e.g., in the context of MTCH1/MTCH2 deletion) is sufficient to promote endolysosomal trafficking of BNIP3 without requiring prior OMM localization. In total, our results support a model where extraneous or mislocalized TA monomers are degraded by the UPS, while dimerization leads to stable protein complexes that are cleared from the ER through trafficking to lysosomes. In such a model, BNIP3 dimers present in the OMM are resistant to both forms of quality control, thus explaining the observed steady-state localization of BNIP3 in the OMM. While other groups have speculated such a model (McKenna et al, 2022, 2020), we provide direct evidence using an endogenous TA protein, BNIP3. Thus, we directly implicate endosomal trafficking and lysosomal degradation in the canon of quality control pathways that support the proper localization of TA membrane proteins.

BNIP3 has been implicated in a variety of physiological processes not considered here (Wrighton et al, 2021; Schmid et al, 2022;

Berardi et al, 2022; Vara-Pérez et al, 2021; Chourasia et al, 2015; Bozi et al, 2018; Zhang et al, 2019; Kothari et al, 2003; Echavarria-Consuegra et al, 2022). Consequently, the full physiological implications of BNIP3 regulation will be an important area of continued study. For instance, a tumor-suppressor function for BNIP3 has been suggested that is independent of its role as a BH3-containing protein (Berardi et al, 2022). Correspondingly, transcriptional repression of BNIP3 is associated with several cancer types (Zhu et al, 2022). Given the extent to which posttranslational mechanisms impinge upon BNIP3 function, we anticipate that posttranslational control of BNIP3 may similarly be exploited by cancerous cells to restrict BNIP3. Since transcriptome-level analyses are blind to this level of regulation, BNIP3's role in tumor progression is likely underestimated.

We note that BNIP3-mediated mitophagy is commonly associated with stress conditions, particularly where hypoxia is a factor as in ischemia/reperfusion injury (Tang et al, 2019; Li et al, 2020; Wrighton et al, 2021). In contrast, NIX generally is implicated in mitophagy during cellular differentiation programs (Ordureau *et al*, 2021; Zhao et al, 2020; Lampert et al, 2019; O'Sullivan et al, 2015; Xiang et al, 2017; Yazdankhah et al, 2021; Sandoval et al, 2008). Future efforts will be needed to further delineate the differential utilization of these highly related mitophagy receptors. However, this utilization trend is generally consistent with the relative responsiveness of BNIP3 and NIX to proteostatic collapse. Going forward, it will be important to fully consider the implications of proteostatic collapse on BNIP3-mediated mitophagy in vivo. For example, deficiencies in EMC have been identified in clinical settings in conditions related to the central nervous system (Harel et al, 2016; Geetha et al, 2017; Abu-Safieh et al, 2013). However, further molecular and cellular characterization will be needed to dissect the interplay of EMC-mediated protein insertion, mitophagy, and neuronal physiology. In addition, bortezomib-induced peripheral neuropathy (BIPN) is a common and painful side effect of bortezomib use as a chemotherapeutic agent (Zheng et al, 2012; Ludman and Mele-medjian, 2019). While the underlying mechanism of BIPN remains a matter of debate, the mitochondrial dysregulation associated with BIPN makes BNIP3-induced mitophagy an intriguing therapeutic candidate.

# Methods

## Antibodies

For IB, all primary antibodies are used at a 1:1000 dilution, unless stated otherwise. Secondary antibodies are used at a 1:10,000 dilution. For immunofluorescence (IF): primary antibodies were diluted 1:100 and secondary antibodies were used 1:1000. The following primary antibodies were used: mouse anti-SQSTM1/p62 (ab56416, Abcam), rabbit anti-NDP52 (9036, CST), rabbit anti-ATG9A (13509S, CST), mouse anti-GFP (118114460001, Sigma), rabbit anti-BNIP3 (44060S, CST), rabbit anti-BNIP3L/NIX (12396S, CST), anti-V5 Tag (13202, CST), rat anti-tubulin (sc-53030, Santa Cruz Biotechnology), mouse anti-tubulin (3873S, CST), and mouse anti-EMC3/TMEM111 (67205-1-lg, Proteintech). The following secondary antibodies were used for (IB): goat anti-mouse IgG(H+L) IRDye 680LT (926-68020, LI-COR), goat anti-rabbit IgG(H+L) IRDye800CW (926-32211, LI-COR); secondary antibodies (IF): goat anti-rabbit IgG(H+L) Alexa Fluor

Plus 647 (A32733, Invitrogen), goat anti-mouse IgG(H+L) Alexa Fluor Plus 647 (A32728, Invitrogen).

## Vectors

The Brunello knockout pooled library was a gift from David Root and John Doench (Addgene #73178). psPAX2 was a gift from Didier Trono (Addgene plasmid # 12260). pCMV-VSV-G was a gift from Bob Weinberg (Addgene plasmid #8454). lentiCRISPRv2puro was a gift from Brett Stringer (Addgene plasmid #98290). lentiGuide-puro was a gift from Feng Zhang (Addgene plasmid #52963). pFUGW-EFSp-Cas9-P2A-Zeo (pAWp30) was a gift from Timothy Lu (Addgene plasmid #73857). pLenti CMV GFP Puro (658-5) was a gift from Eric Campeau & Paul Kaufman (Addgene plasmid #17448). mitoBFP was a gift from Gia Voeltz (Addgene # 49151). pGW1-mCherry-EGFP-PIM was a gift from Lukas Kapitein (Addgene plasmid #111759). pHAGE-mt-mKeima was a gift from Richard Youle (Addgene plasmid #131626). pLKO.1 hygro was a gift from Bob Weinberg (Addgene plasmid # 24150) Other vectors generated during this study are available upon request.

## Chemicals and reagents

The chemicals and reagents used in this study include: 2-Deoxy-D-glucose (D8375-1G, Sigma), 2-mercaptoethanol (BME) (M6250-100ML, Sigma), agar (A10752, AlphaAesar), agarose (16500500, Thermo Fisher), ampicillin (A9518-25G, Sigma), Bafilomycin-A1 (11038, Caymen Chemical), Beta-glycerophosphate (35675-50GM, Sigma), blasticidin (ant-bl-1, Invivogen), Bortezomib (10008822, Caymen Chemical), Brefeldin-A1 (11861, Caymen Chemical), CB-5083 (19311, Caymen Chemical), dimethyl sulfoxide (C833V25, Thomas Scientific), EDTA (EDS-500G, Sigma), glycerol (G2025-1L, Sigma), HEPES (H3375-1KG, Sigma), hygromycin (ant-hg-1, Invivogen), kanamycin (BP906-5, FisherSci), MLN-4924 (15217, Caymen Chemical), MLN-7243 (30108, Caymen Chemical), plasmocin (ant-mpp, Invivogen), Phusion High-Fidelity DNA polymerase (M0530L, NEB), PIK-III (17002, Caymen Chemical), polybrene (H9268-5G, Sigma), potassium chloride (P217-500, FisherSci), puromycin (ant-pr-1, Invivogen), sodium chloride (6438, FisherSci), sodium deoxycholate (97062-028, VWR), sodium dodecyl sulfate (SDS) (74255-250G, Sigma), sodium fluoride (S6776-100G, Sigma), sodium orthovanadate (450243-10G, Sigma), sodium pyrophosphate decahydrate (221368-100G, Sigma), Taq DNA ligase (M0208L, NEB), TERGITOL Type NP-40 (NP40S-100ML, Sigma), Tris base (T1378-5KG, Sigma), Triton X-100 (T9284-500ML, Sigma), tryptone (DF0123-17-3, FisherSci), Tween-20 (BP337-500, FisherSci), T5 exonuclease (M0363S, NEB), yeast extract (BP1422-2, FisherSci), and zeocin (ant-zn-1, Invivogen).

## Tissue culture

All mammalian cells were grown in a standard water-jacketed incubator with 5% $CO_2$. MDA-MB-231, U2OS, HEK293T, MDA-MB-453 all grown in DMEM media (45000-304, Corning) with 10% FBS (26140079, Gibco) and 1× penicillin/streptomycin (15140122, Thermo Scientific). K562 cells were grown in IMDM media (45000-366, Corning) with 10% FBS and 1× penicillin/streptomycin. Mammalian cells (HTB-26, MDA-MB-231; CCL-243, K562;

CRL-3216, HEK293T) were acquired from the American Type Culture Collection (ATCC) where cell authentication is performed. Cell authentication was not performed on U2OS and MDA-MB-453 cells. Plasmocin prophylaxis (1:500) was used when generating of new stable cell line. All cells were maintained below an 85% confluency and passaged less than 25 times. For passaging, cells are trypsinized with 0.25% Trypsin-EDTA (25200114, Thermo Scientific). For hypoxia incubation, cells were incubated in a humidified Baker Ruskinn InvivO$_2$ (I400) hypoxia chamber at 1% O$_2$ and 5% CO$_2$ for the indicated times. Puromycin (2 µg/mL), blasticidin (5 µg/mL), and zeocin (50 µg/mL) were added when necessary for selection. In all, 1× Hanks' Balanced Salt Solution (HBSS) (45000-456, Corning) was used to wash cells when passaged.

## Generation of gene knockout cell line

Sequences for sgRNAs were generated from the Brunello library and cloned into the indicated vectors as outlined above under "sgRNA oligonucleotide ligation protocol". HEK293T and MDA-MB-231 cells were transfected with the resulting vectors. Single-cell sorting was used to isolate individual clones. Clonal knockouts were confirmed by immunoblotting (IB).

## Molecular cloning and bacterial transformation

PCR products were amplified using Phusion High-Fidelity DNA polymerase (M0530L, NEB). Amplification primers were designed with a 30-base pair overlap with the recipient vectors. Linearized vector backbones were dephosphorylated by calf intestinal phosphatase (M0290S, NEB). All inserts and vectors were purified from a 1% agarose gel prior to isothermal assembly (D4002, Zymo Research). In total, 20 ng of linearized vector DNA was combined with isomolar amounts of purified insert(s). Overall, 2.5 µL DNA mix was incubated with 7.5 µL isothermal assembly master mix at 50 °C for 20 min. The product of the isothermal assembly reaction was transformed into NEB Stable cells (C3040H, NEB). Transformed cells were plated on plates of LB media (10 g/L tryptone, 5 g/L yeast extract, 5 g/L NaCl) containing 1.5% agar, 100 µg/mL ampicillin or 50 µg/mL kanamycin were included in bacterial cultures. All plates were grown overnight at 34 °C and transformed colonies were grown overnight at 34 °C. Overnight cultures were pelleted at 3000 × g for 10 min and plasmid DNA was purified using a Qiagen miniprep kit (27106, Qiagen). Sequences were verified by Sanger sequencing (Eton Bioscience Inc).

## sgRNA oligonucleotide ligation

Oligonucleotides were ordered from Thermo Fisher. For sgRNA cloning, oligos were ordered in the following format: forward: 5'-CACCGNNNNNNNNNNNNNNNNNNNN-3'; reverse: 5'-AAAC NNNNNNNNNNNNNNNNNNNNC-3'. For shRNA cloning, oligos were ordered in the following format: forward: 5'-CCGGNx 48TTTTTG-3'; reverse: 5'-AATTCAAAAANx48-3'. In total, 50 pmol of each oligo was mixed in a 25 µL reaction and phosphorylated with T4 polynucleotide kinase (M0201S, NEB). Reactions were performed for 30 min at 37 °C in 1× T4 DNA ligase buffer (B0202S, NEB). Phosphorylated oligos were annealed by heating for 5 min at 95 °C and slow cooling (0.1 °C/s). In total, 2 µl of diluted (1:100) oligo mix was ligated into 20 ng BsmBI-digested

vector (pLentiGuide-puro or pLenti-CRIPSR v2), or AgeI/EcoRI-digested vector (pLKO.1 hygro for shRNA), using T4 DNA ligase (M0202S, NEB). The ligation reaction was done at room temperature for at least 15 min prior to bacterial transformation. All small hairpin and sgRNA sequences are listed in Table EV1.

## Flow cytometry

Cells were trypsinized, centrifuged, and resuspended in cold 1× HBSS and filtered through a 41-µm nylon mesh prior to flow analysis. All flow cytometry data was collected on a Beckman Coulter CytoFLEX flow cytometer. Data was analyzed using FlowJo v10 and R Studio. At least 10,000 cells were collected for all samples.

## Lentiviral generation

Lentivirus was generated using HEK293T cells with Lipofectamine 3000 kit (L3000008, Life Technologies). Cells were seeded in Opti-MEM media, containing 5% FBS and no antibiotics, overnight for ~80% confluency. Cells were transfected with packaging vectors pVSV-G and pSPAX2, along with expression construct at a 1:4:3 ratio, scaled accordingly. Opti-MEM media was exchanged with fresh media 6 h after transfection. The supernatant containing virus was collected at 24- and 48 h post-transfection and pooled together. The virus was cleared by centrifugation for 15 min at 1000 × g and aliquoted to avoid freeze–thaw cycles.

## Viral transduction

Cells were incubated in respective media containing 8 µg/mL polybrene (1:1000 dilution) with the virus. If adherent, cells were tryspinized and allowed to re-adhere with media containing polybrene and virus. Transduction were left overnight, and virus-containing media was exchanged in the morning with fresh media lacking polybrene. Transduced cells were allowed to recover in fresh media for 24 h prior to antibiotic selection.

## Transient transfection

Cells were seeded at ~75% confluency in Opti-MEM reduced media supplemented with 5% FBS no antibiotics and allowed to adhere overnight. Cells were transfected using Lipofectamine 3000 reagent (L3000008, Life Technologies), according to the manufacturer's protocol. Cells were left in a Lipofectamine reaction for 6 h before a fresh Opti-MEM media exchange. Cells were analyzed 24 h post-transfection.

## Gel electrophoresis and Immunoblotting

Cells are harvested from respective dishes or wells. Cells were centrifuged to remove the media. Cells are resuspended and washed once in 1× HBSS prior to lysis. Cells are lysed for 20 min on ice in lysis buffer (50 mM HEPES pH 7.4, 40 mM NaCl, 2 mM EDTA, 1% Triton X-100, 2× complete protease inhibitor tablet (5056489001, Sigma)). Lysates were cleared by centrifugation for 8 min at 1000 × g, using supernatants as sample input. Total protein level was normalized using a BCA protein assay (23225, Thermo Scientific) and adjusted with lysis buffer. Normalized lysate samples were denatured at 65° for 10 min in 1× (final concentration) Laemmli Loading Buffer (3× stock: 189 mM Tris pH 6.8, 30%

glycerol, 6% SDS, 10% beta-mercaptoethanol, bromophenol blue). Gel electrophoresis was performed at 175V for 60 min in Novex 4–20% Tris-Glycine gels. Protein gels were transferred for 60 min to 0.2-μm PVDF membranes (ISEQ00010, Sigma) using the semi-dry Trans-Blot Turbo Transfer system (Bio-Rad). Membranes were blocked for at least 30 min in 5% milk in 1× TBST (MP290288705, Fisher Scientific). Primary antibodies were diluted in 5% milk in TBST and incubated on membrane overnight at 4 °C. After overnight primary incubation, membranes were washed three times in 1× TBST for 5 min. Secondary antibodies were diluted in Intercept™ (TBS) Blocking Buffer (927-60003, LI-COR) and incubated on the membrane for 1 h at room temperature. After secondary incubation, membranes were washed twice in 1× TBST and last wash was done in 1× TBS (no Tween). All membranes were imaged on LI-COR Odyssey CLx dual-color imager and band intensities were quantified on LI-COR analysis software Image Studio Lite.

## Mito-Keima assays

For BFP-BNIP3 overexpression, MDA-MB-231 cells stably expressing mt-Keima reporter were transduced following normal viral transduction. Transduced cells were either treated with vehicle (DMSO) or Baf-A1 (100 nM) after 24 h post transduction for 18 h and analyzed by flow cytometry 48 h post transduction. For drug treatment, MDA-MB-231 cells stably expressing mt-Keima reporter were treated with the respective drug for 18 to 24-h prior to flow cytometry. Baf-treated and non-transduced samples served as gating controls for all mt-Keima flow analysis.

## Measuring oxygen consumption

Oxygen consumption and glycolytic rates were analyzed using the Seahorse XF96 system. Cells were seeded on Seahorse XF96 cell culture microplates (101085-004, Agilent) at a density of $0.25 \times 10^5$ density per well in DMEM media supplemented with 10% FBS and no antibiotic selection 24 h prior to analysis. DMEM media was exchanged with Mito Stress XF DMEM media and incubated for 35 min prior to test. The Mito Stress Test (103015-100, Agilent) was performed the following day, using the manufacturer's protocol (Injection 1: Oligomycin 1.5 μM; Injection 2: FCCP 1 μM; Injection 3: Rotenone/Antimycin A 0.5 μM; Injection 4: 2-deoxy-D-glucose 50 mM). Respiration and glycolytic rates were calculated based on the manufacturer's protocol. The Seahorse XF96 analyzer from the Immune Monitoring and Flow Cytometry Shared Resource (DartLab) was used. All Seahorse data was normalized by cellular lysis using RIPA lysis buffer (150 mM NaCl, 50 mM Tris-HCl pH 8, 0.5% sodium deoxycholate, 0.1% SDS, 1% TERGITOL Type NP-40 solution, 2× complete protease inhibitor tablet) in the microplate and performing a BCA protein assay.

## Artificial dimerization assay

The DmrB inducible dimer domain was subcloned from pGW1-mCherry-EGFP-PIM (Addgene #111759) to the N-terminal cytosolic portion of BNIP3. DmrB constructs were packaged in lentivirus and transduced into cells. Transduced cells were exchanged with fresh media and allowed to recover for 24 h. On day 2 post transduction, cells were exchanged with fresh media containing B/B homodimerizer (500 nM) (Takara Bio, #635059) and/or Baf-A1 (100 nM) and/or vehicle (DMSO) control for 6 h prior to flow cytometry analysis.

## In vitro dephosphorylation assay

Cells were transduced with lentivirus for expression of V5-BNIP3 variants. Cells were lysed in high salt/IP lysis buffer (50mM HEPES pH 7.4, 150 mM NaCl, 2 mM EDTA, 1% Triton X-100, 2× complete protease inhibitor tablet (5056489001, Sigma)). Lysates were cleared by centrifugation. In total, 25 μL of Magnetic V5-Trap bead slurry (v5tma-10, Chromotek), per lysate sample, was washed once with IP lysis buffer and incubated with cleared lysates for 40 min at 4 °C on end-over-end rotator. Pull-down flow-through was collected after bead-lysate incubation. Bead slurry was divided into three and washed five times with IP lysis buffer. All three bead samples were moved to PMP buffer (P0753S, NEB), corresponding to PMP buffer only, PMP with Lambda Protein Phosphatase, and PMP with Lambda Protein Phosphatase (P0753S, NEB) with Phosphatase inhibitor cocktail (4×: 5 mM NaF, 1 mM orthovanadate, 1 mM pyrophosphate, 1 mM glycerophosphate). For 50 μL reactions, the following volumes were used: 5 μL of 10× PMP buffer, 5 μL of MnCl₂ (10 mM), 0.75 μL of Lambda Protein Phosphatase, and 4× Phosphatase inhibitor cocktail. Dephosphorylation reaction was done for 30 min at 30 °C prior to sample denaturing.

## Immunofluorescence and live-cell microscopy

Cells were seeded on glass bottom dishes (07-000-235 and NC0832919, Fisher Scientific) at approximately 70% confluency and allowed to adhere overnight. Cells were washed twice in warm 1X HBSS and fixed for 15 min in 4% paraformaldehyde (PFA) made from fresh 16% PFA (#15710, Electron Microscopy Sciences) diluted with 1× HBSS. Cells were blocked and permeabilized at room temperature for 1 h in Intercept™ (TBS) Blocking Buffer (927-60003, LI-COR) plus 0.3% Triton X-100, then washed once in 1× HBSS. Primary antibody was diluted in (1:100), and cells were incubated in Intercept™ (TBS) Blocking Buffer of primary antibody solution overnight at 4 °C. After incubation, cells were washed 3 × 5 min in 1× HBSS. The secondary antibody was diluted to 1:1000 in Intercept™ (TBS) Blocking Buffer, and cells were incubated in secondary antibody solution for 45 min at room temperature. After incubation, cells were washed 3 × 10 min in 1× HBSS, stained with a 1:10,000 dilution of Hoechst 33342 (H3570, Thermo Fisher) for 5min, and washed once more in 1× HBSS before storage or imaging. For living cell imaging, cells were seeded on glass bottom dishes at ~70% confluency and imaged in DMEM with no phenol red (21-063-029, Fisher Scientific) supplemented with 10% FBS.

## Confocal microscopy

All fluorescent images were obtained using a Nikon Eclipse Ti-E inverted microscope stand that has a Yokogawa, two-camera, CSU-W1 spinning disk system with a Nikon LU-N4 laser launch. All images were obtained on ×100 PlanAPO objective lens.

## Image analysis

Image intensities were modified linearly and evenly across samples per experiment. All images represent a single plane on the acquired Z-stack and processed in ImageJ. For co-localization analysis, the Coloc2 plugin on ImageJ was used. Image intensities were modified and the threshold for the two channels of interest. Cellular segmentation in images was done manually to obtain region of interests (ROIs). ROIs underwent Coloc2 analysis.

## Library lentiviral generation

Brunello Library was purchased from Addgene (#73178). Lentivirus for Brunello library was generated by lipofection of HEK293T cells with 5 µg psPAX2, 1.33 µg pCMV-VSV-G, and 4 µg of library vector per 10-cm plate of HEK293T at 85% confluency. Freshly pulled HEK293T cells were grown in Opti-MEM containing 5% FBS and no antibiotics. Opti-MEM media was exchanged 6 h after transfection. Supernatants containing virus were collected at 24 h post-transfection, replenished, and collected again at 48 h. Supernatants were pooled and cleared by centrifugation for 15 min at $1000 \times g$. Viral titer was quantified using the Lenti-X™ qRT-PCR Titration Kit (Takara Bio, #631235), according to the manufacturer's protocol.

## Transduction and cell growth

For CRISPR screening experiments, MDA-MB-231 cells were passaged to maintain cell density between 80–90% confluency in 10-cm dishes. Cells were propagated in DMEM+10% FBS + pen/strep + appropriate antibiotics (Blasticidin 5 µg/ml, zeocin 50 µg/ml) until 100 million cells were obtained. Hundred million cells were trypsinized and resuspended in DMEM + 10% FBS + 8 µg/mL polybrene. An MOI of 0.4 was used to minimize multiple infection events per cell. The date of infection was day 0. Cells were infected overnight and exchanged into fresh media. After 24 h, 2 µg/ml puromycin was added. Cells were expanded to 15-cm dishes in puromycin. Cells were removed from puromycin 1 day prior to sorting and on days 8, 11, and 12, cells were sorted for red:green fluorescence, sorting 200 million cells each day. In total, 50 million unsorted cells were collected and processed as input. The top and bottom 30% of cells (based on Red:Green ratio) were taken. Cell sorting was performed using a Sony SH800 cell sorter. Cells were pelleted and stored at $-80\,°C$ until processing.

## CRISPR screen processing

Genomic DNA was purified from collected cells using the NucleoSpin Blood XL kit (740950.1, Macherey Nagel) according to the manufacturer's instructions. Brunello library sgRNA sequences were amplified from total genomic DNA using a common pool of eight staggered-length forward primers. Unique 6-mer barcodes within each reverse primer allowed multiplexing of samples. Each 50 µL PCR reaction contained 0.4 µM of each forward and reverse primer mix (Integrated DNA Technologies), 1× Phusion HF Reaction Buffer (NEB), 0.2 mM dNTPs (NEB), 40 U/ml Phusion HF DNA Polymerase (NEB), up to 5 lg of genomic DNA, and 3% v/v DMSO. The following PCR cycling conditions were used: $1 \times 98\,°C$ for 30 s; $25 \times$ ($98\,°C$ for 30 s, $56\,°C$ for 30 s, $63\,°C$ for 30 s); $1 \times 63\,°C$ for 10 min. The resulting products were pooled to obtain the sgRNA libraries. The pooled PCR

products were size selected between 0.60× and 0.85× magnetic bead slurry. Library purity and size distribution was measured on a Fragment Analyzer instrument (Agilent) and quantified fluorometrically by Qubit. Libraries were pooled in equimolar ratios and loaded at 2.5 pM on to a NextSeq500 High Output 75-cycle run. 2% PhiX spike in was included as an internal control for sequencing run performance. Data were demultiplexed into fastq files using Illumina bcl2fastq2v2.20.0.422. All fastq files are available on Mendeley Data.

## NGS data analysis

The 5' end of Illumina sequencing reads was trimmed to 5'-CACCG-3' using the Cutadapt feature (Martin, 2011). The count function of MAGeCK (version 0.5.9) was used to extract read counts for each sgRNA (Li et al, 2014). Trimmed fastq files are available on Mendeley Data. The RRA function was used to compare read counts from cells displaying increased and decreased Red:Green ratios (Li et al, 2014). The output included fold change, rank, and *P* value. Average fold change scores (across three experimental replicates), and *P* values can be found in Dataset EV1.

## Statistical analysis

All statistical analysis was performed using Prism 8 (GraphPad). All statistical tests are indicated in the relevant figure legends. All tests were two-tailed with *P*<0.05 as the threshold for statistical significance. The number of replicates (*n*) used for each experiment and statistical test are indicated in the relevant figure legend.

# Data availability

All source data can be found on Mendeley Data for anonymous public access including raw fcs files from flow cytometry, trimmed fastq files from CRISPR screen, and acquired nd2 files from microscopy. Table Expanded View 2 (Table EV2) contains all DOIs for source data under the Mendeley Data repository.

# Peer review information

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

## Acknowledgements

We would like to thank former lab member Amelia Ohnstad for technical assistance. We would like to thank other members of the Shoemaker Lab for moral support. We would like to thank Dr. Michael Ragusa and lab members for providing insightful feedback. We would like to thank Dr. Robert Cramer and Kaesi Morelli for assistance with the hypoxia chamber. We would like to thank Vladimir Denic, Michael Ragusa, and Charles Barlowe for the critical reading of the manuscript. This work is supported by the National Institutes of Health General Medical Sciences (R35GM142644 to CJS, F31GM143849 to JMD). We would like to thank Ann Lavanway for light microscopy expertise. We would like to thank the Institute for Biomolecular Targeting (bioMT) core at Dartmouth supported by National Institutes of Health General Medical Sciences (P20GM113132). We would like to thank the Genomics Shared Resource and the Immune Monitoring and Flow Cytometry Shared Resource (DartLab) supported by National Cancer Institute (P30CA023108) to the Dartmouth Cancer Center.

## Author contributions

**Jose M Delgado**: Conceptualization; Resources; Data curation; Software; Formal analysis; Supervision; Funding acquisition; Validation; Investigation; Visualization; Methodology; Writing—original draft; Project administration; Writing—review and editing. **Logan Wallace Shepard**: Conceptualization; Data curation; Formal analysis. **Sarah W Lamson**: Data curation. **Samantha L Liu**: Data curation. **Christopher J Shoemaker**: Conceptualization; Resources; Data curation; Software; Formal analysis; Supervision; Funding acquisition;

Validation; Investigation; Visualization; Methodology; Writing—original draft; Project administration; Writing—review and editing.

## Disclosure and competing interests statement

The authors declare no competing interests.

# Expanded View Figures

▶

**Figure EV1.   Related to Fig. 1.**

(**A**) MDA-MB-231 cells were transduced with V5-BNIP3 variants and lysed 48 h post transduction. The V5 epitope was immunoprecipitated from extracts and treated with buffer alone (lane 1), lambda phosphatase (PP, lane 2), or lambda phosphatase with phosphatase inhibitor cocktail (PIC, lane 3). (**B**) Immunoblotting of MDA-MB-231-derived extracts from wild-type (WT) and $ATG9^{KO}$ clonal knockout cells. Where indicated, cells were treated with Baf-A1 (100 nM) for 18 h. (**C–E**) Immunoblotting of MDA-MB-231, K562, U2OS, MDA-MB-453-derived extracts from cells expressing Cas9 and the indicated sgRNA. Cells were subjected to normoxia or hypoxia and/or Baf-A1 treatment (100 nM) for 18 h where indicated. (**F**) Immunoblotting of extracts derived from parental HEK293T and clonal $ATG9^{KO}$ knockout cells. Where indicated, cells were treated with Baf-A1 (100 nM) for 18 h. (**G**) Violin plots of MDA-MB-231 cells expressing either the tf-NDP52 or tf-BNIP3 reporter. Cells were treated with DMSO or Baf-A1 (100nM) or PIK-III (10 μM) for 18 h before being analyzed by flow cytometry for red:green ratio. Median values for each sample are identified by a black line within each violin. The red dotted line across all samples corresponds to red:green ratio of maximally inhibited conditions (Baf-A1) (*n* >10,000 cells). (**H**) Violin plots of MDA-MB-231 cells expressing tf-BNIP3 transduced with either a control small hairpin RNA (shCtrl) or an shRNA targeting Rab7 (shRab7). Cells were analyzed for red:green ratio 8 days post transduction. Red dotted line (=1) corresponds to the theoretical maximum inhibition of red:green ratio. (*n* >10,000 cells). (**I**) Quantification of Pearson's correlation coefficients from cells in Fig. 1D. Correlation of RFP to mitoBFP (reflective of mitochondrial localization) was calculated using Coloc2. Bar graphs represent mean $+/-$ SEM. Each data point represents a single cell. *n* = 15 cells. Statistical analysis was performed using an unpaired Student's *t* test. *$P$<0.05.

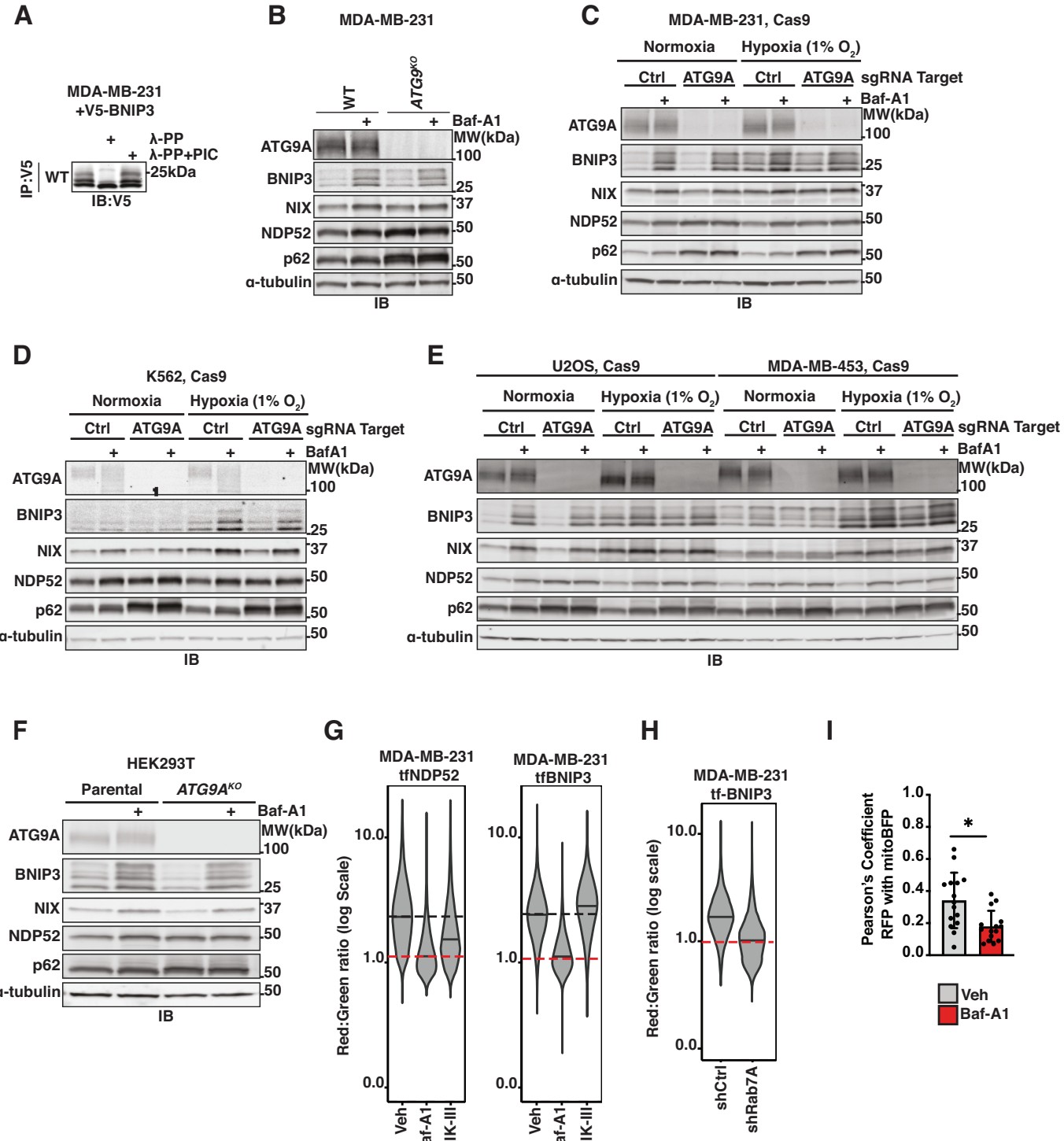

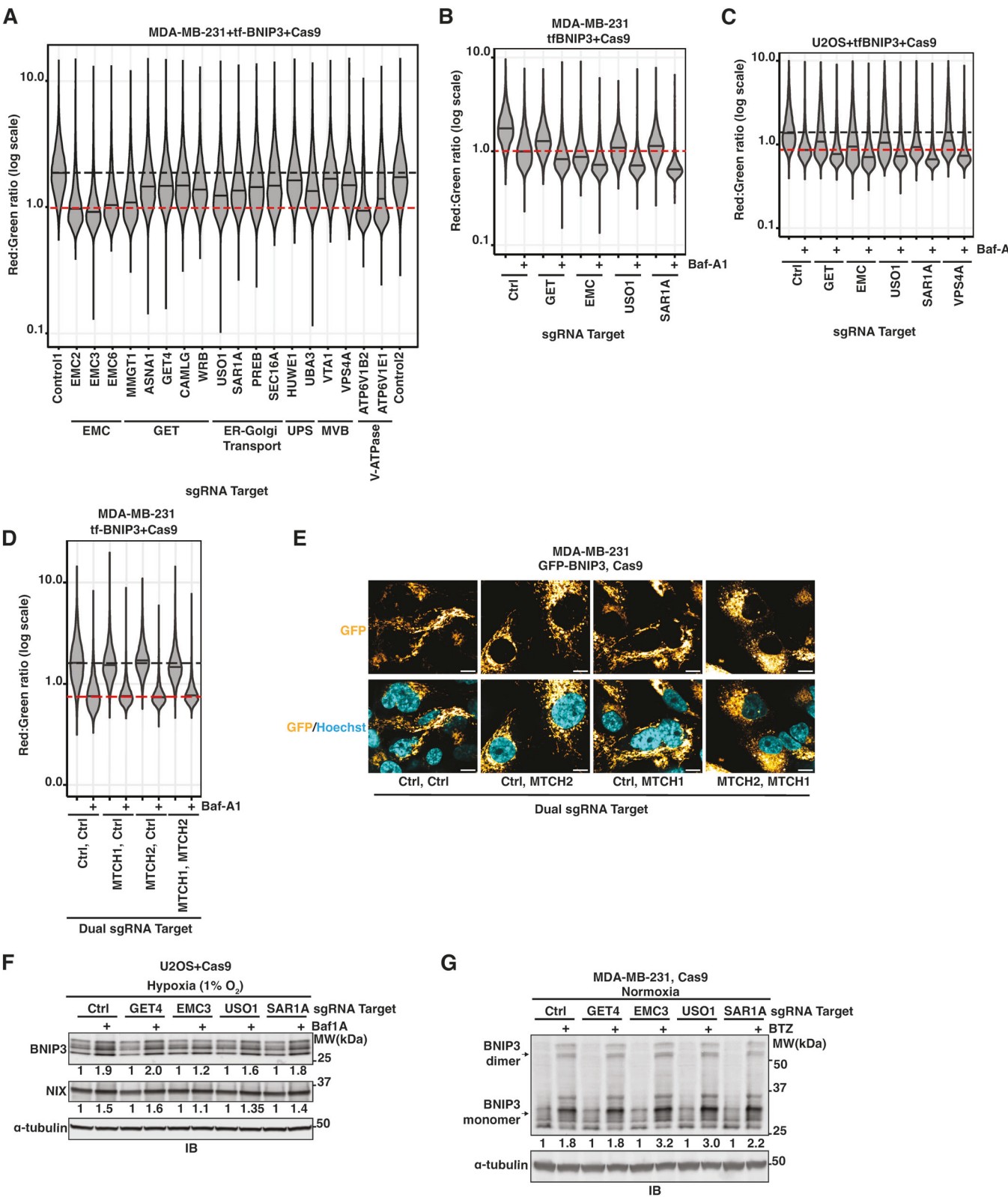

◀  **Figure EV2.   Related to Fig. 3.**

(A) MDA-MB-231 cells expressing tf-BNIP3 and Cas9 were transduced with the indicated sgRNAs. Red:green ratio was analyzed by flow cytometry on day 8 post transduction. Median values for a nontargeting control (sgControl1) are identified by a dashed black line. The red dotted line across all samples corresponds to a red:green ratio of 1, the theoretical minimum ($n$ >10,000 cells). (B) MDA-MB-231 cells expressing tf-BNIP3 and Cas9 were transduced with the indicated sgRNAs. Red:green ratio was analyzed by flow cytometry on day 8 post transduction. The red dotted line across all samples corresponds to red:green ratio of Baf-A1-treated control (Ctrl) cells ($n$ >10,000 cells). (C) U2OS cells expressing tf-BNIP3 and Cas9 were transduced with the indicated sgRNAs. Red:green ratio was analyzed by flow cytometry on day 8 post transduction. The red dotted line across all samples corresponds to red:green ratio of Baf-A1-treated control (Ctrl) cells ($n$ >10,000 cells). (D) MDA-MB-231 cells expressing tf-BNIP3 and Cas9 were transduced with the indicated dual sgRNAs. Red:green ratio was analyzed by flow cytometry on day 8 post transduction. The red dotted line across all samples corresponds to red:green ratio of Baf-A1-treated control (Ctrl) cells ($n$ > 10,000 cells). (E) Representative confocal micrographs of MDA-MB-231 cells expressing GFP-BNIP3 and Cas9. Cells were transduced with indicated sgRNAs, propagated for 8 days and fixed for image aquisition. Hoechst stain was used for nuclear staining. Scale bar is 10 μm. (F) Immunoblotting of U2OS-derived extracts expressing Cas9 that were transduced with the indicated sgRNAs. On day 8 post transduction, cells were treated with Baf-A1 (100 nM) and subjected to hypoxia for 18 h prior to lysis. (G) Immunoblotting of MDA-MB-231-derived extracts expressing Cas9 that were transduced with the indicated sgRNAs. Cells were treated with Bortezomib (100 nM) for 18 h on day 8 post transduction, prior to lysis.

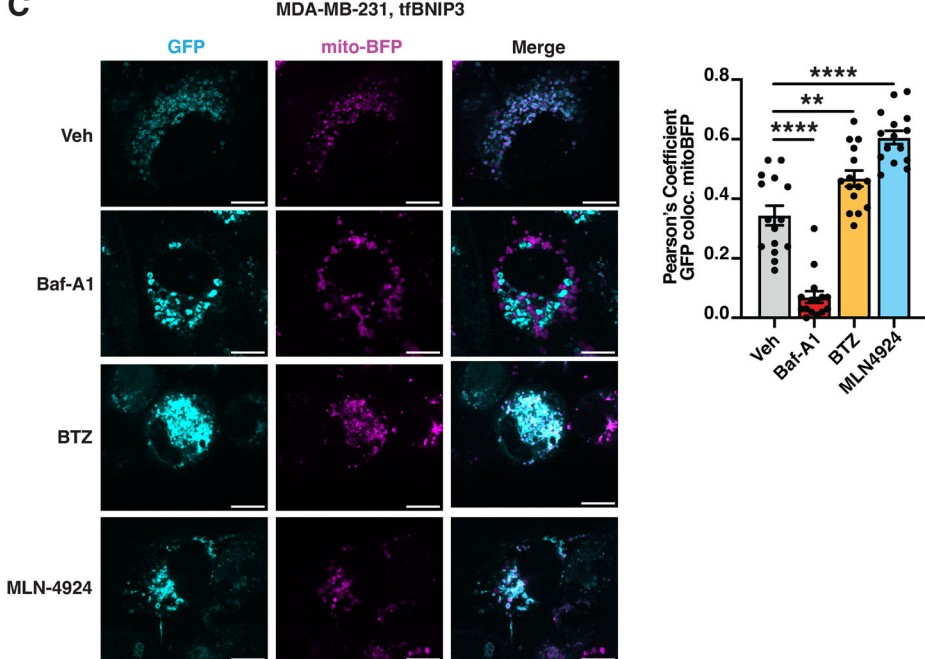

**Figure EV3.   Related to Fig. 4.**

(**A**) Representative image of one biological replicate quantified in Fig. 4A. Immunoblotting (IB) of MDA-MB-231-derived extracts from cells treated with vehicle (DMSO), Baf-A1 (100 nM), MLN-4924 (1 μM), CB-5083 (1 μM) and subjected to hypoxia for 18 h. (**B**) Quantification of protein accumulation from Figs. 4D and EV3A. Bar graphs represent mean $+/-$ SEM from four independent experiments. All protein levels were normalized to α-tubulin. Statistical analysis was performed using a one-sample *t* test to the normalized control. \*\**P* < 0.01; \**P* < 0.05; ns not significant. (**C**) Representative confocal micrographs of tf-BNIP3-expressing cells treated with Baf-A1 (100 nM), MLN-4924 (1 μM), or Bortezomib (100 nM) for 18 h. Pearson's correlation coefficient between GFP and mitoBFP (reflective of mitochondrial localization) was calculated using Coloc2. Bar graphs represent mean $+/-$ SEM. Each data point represents a single cell. Statistical analysis was performed using an unpaired t test. Scale bar: 10 μm; *n* = 15 cells; \*\*\*\**P* < 0.0001, \*\**P* < 0.01.

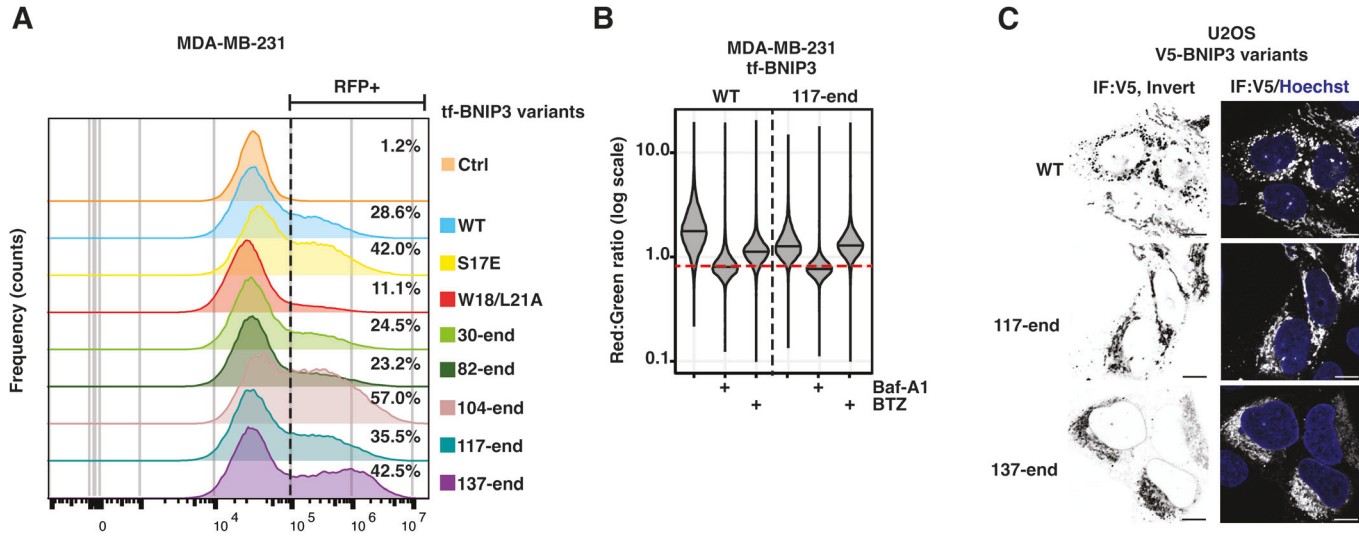

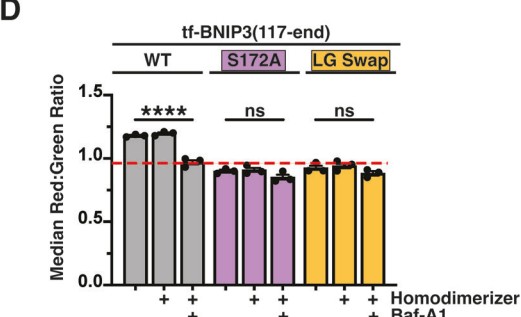

**Figure EV4. Related to Fig. 5.**

(A) Histograms for expression of tf-BNIP3 variants. MDA-MB-231 cells were transduced with the indicated tf-BNIP3 variants and analyzed by flow cytometry 48 h post transduction. Threshold for mRFP+ cells was determined by a non-transduced control (Ctrl). The percentage of RFP+ cells are indicated for each sample. (*n* > 10,000 cells). (B) MDA-MB-231 cells were transduced with the indicated tf-BNIP3 variants. Red:green ratio was analyzed by flow cytometry 48 h post transduction. Cells were treated with vehicle (DMSO), Baf-A1 (100 nM), or BTZ (100 nM) for 18 h prior to performing flow cytometry. The red dotted line across each sample group corresponds to the maximum inhibition red:green ratio of the wild-type (WT) Baf-A1-treated sample (*n* > 10,000 cells). (C) Representative confocal micrographs of U2OS cells transduced with V5-BNIP3 variants. 48 h post transduction, cells were fixed and immunostained for the V5 epitope. Hoechst stain was used for nuclear staining. Scale bar is 10 μm. (D) MDA-MB-231 cells were transduced with the indicated tf-BNIP3^117-end variants. Red:green ratio was analyzed by flow cytometry 48h post transduction. Cells were treated with Baf-A1 (100 nM) and/or B/B homodimerizer (0.5 μM) for 6 h prior to performing flow cytometry. Bar graphs represent mean +/− SEM from three independent experiments. The red dotted line across all samples corresponds to cells inhibited with Baf-A1 (*n* > 10,000 cells). Statistical analysis was performed using a two-way ANOVA with Dunnett's test. ****$P$<0.0001.

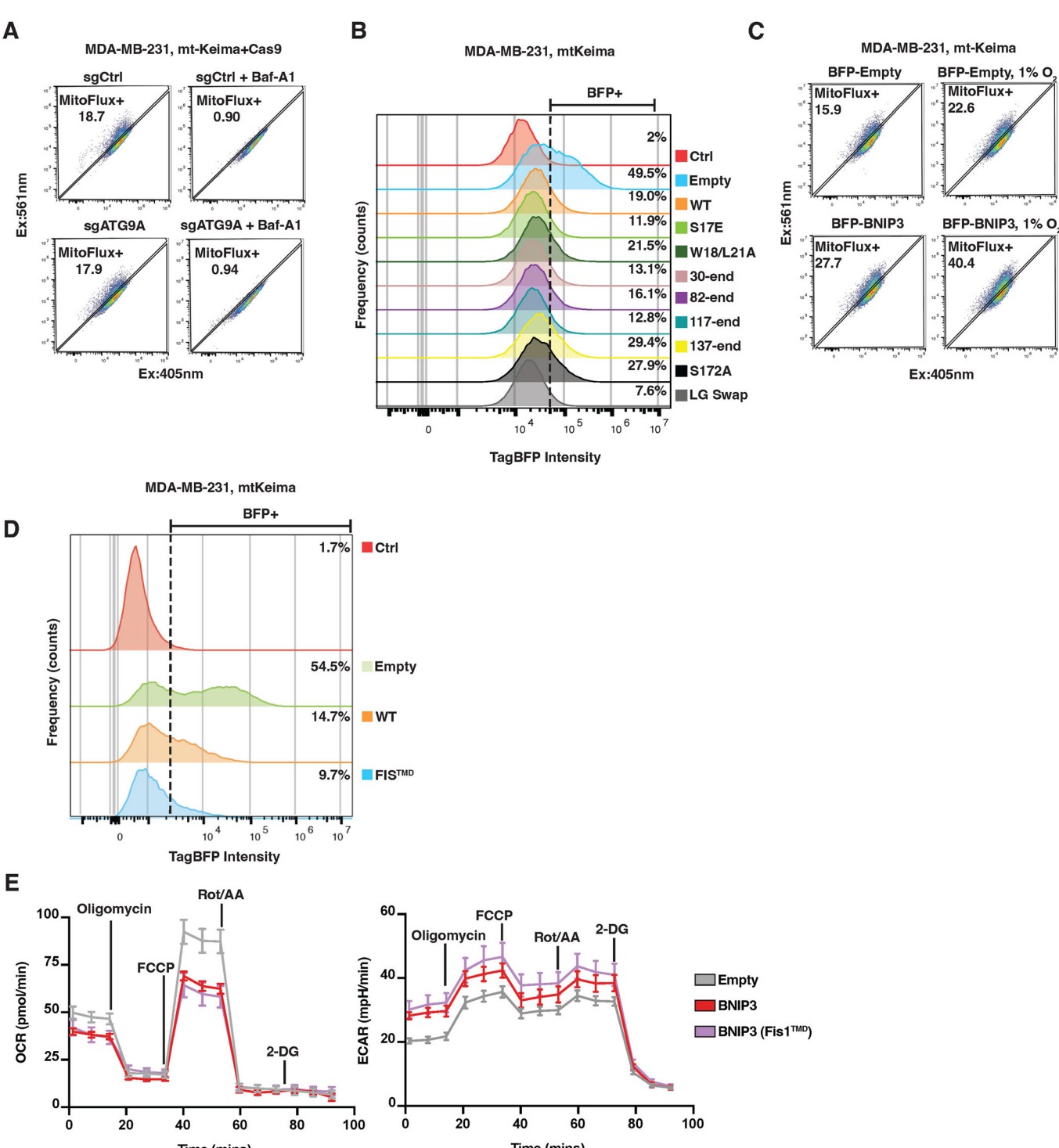

**Figure EV5.   Related to Fig. 6.**

(A) MDA-MB-231 cells expressing mt-Keima were transduced with either a nontargeting sgRNA (sgCtrl) or sgATG9A. On day 8 post transduction, cells were incubated with vehicle (DMSO) or Baf-A1 (100 nM) for 18 h and assessed by flow cytometry. (n > 10,000 cells). (B) Histograms for expression of BFP-BNIP3 variants. MDA-MB-231 mt-Keima cells were transduced with the indicated TagBFP-BNIP3 variants and analyzed by flow cytometry 48 h post transduction. Threshold for TagBFP+ cells was determined by a non-transduced control (Ctrl). The percentage of TagBFP+ cells are indicated for each sample. (n > 10,000 cells). (C) MDA-MB-231 cells expressing mt-Keima were transduced with BFP-BNIP3. At 24 h post transduction, cells were incubated in normoxic or hypoxic conditions for 18 h and assessed by flow cytometry. (n > 10,000 cells). (D) Histograms for expression of BFP-BNIP3 variants. MDA-MB-231 mt-Keima cells were transduced with the indicated TagBFP-BNIP3 variants and analyzed by flow cytometry 48 h post transduction. Threshold for TagBFP+ cells was determined by a non-transduced control (Ctrl). Percentage of TagBFP+ cells are indicated for each sample. (n > 10,000 cells). (E) MDA-MB-231 cells were transduced with indicated the BFP-BNIP3 variants and analyzed for oxygen consumption rate (OCR) and extracellular acidification rate (ECAR) 48 h post transduction. Values were normalized by BCA protein assay. Graphs represent the mean +/− SEM from five technical replicates.

