## [Peer Review File · The EMBO Journal]

The ER membrane protein complex restricts mitophagy by controlling BNIP3 turnover

Jose Delgado, Logan Wallace Shepard, Sarah Lamson, Samantha Liu, and Christopher Shoemaker
DOI: N/A

Corresponding author: Christopher Shoemaker (Christopher.J.Shoemaker@Dartmouth.edu)

Review Timeline:

Transferred from Review Commons:	7th Jun 23
Editorial Decision:	17th Jun 23
Revision Received:	15th Sep 23
Editorial Decision:	20th Oct 23
Revision Received:	1st Nov 23
Accepted:	8th Nov 23

Editor: Daniel Klimmeck

Transaction Report:

Review
COMMONS

This manuscript was transferred to The EMBO Journal following peer review at Review Commons.

Review #1

1. Evidence, reproducibility and clarity:

Evidence, reproducibility and clarity (Required)

Recent work by several groups has revealed that NIX and BNIP3 levels can be regulated through ubiquitination, mediated by FBXL4, to restrict mitophagy. In this study, the authors identify an additional avenue for the regulation of BNIP3 levels involving the transfer BNIP3 from mitochondria to the ER and eventually into the endolysosomal system for degradation. In addition, the authors argue that most of BNIP3 turnover during mitophagy is through the newly identified ER pathway and not through mitophagy. There was little to no endolysosomal turnover observed for NIX, and therefore the authors predominantly focused on BNIP3. Key ER transfer factors required for BNIP3 endolysosomal turnover were identified through whole genome CRISPR/Cas screening, and include EMC3. Knockout of EMC3 results in slightly higher levels of mitophagy under basal conditions, and higher levels of mitophagy following proteasome inhibition with BTZ, supporting the overall conclusion that BNIP3 levels are regulated by lysosomal turnover.

Major Comments:

1. Across the manuscript, NIX levels appear to be unresponsive to most treatments in the MDA-MB-231 line, including hypoxia treatment. This is an unusual result and raises questions about the role of NIX in MDA-MB-231 line, mainly that BNIP3 is the primary driver of mitophagy in this system. Indeed, Figure 7D indicates that there is very little mitophagy contribution by NIX since knockout of BNIP3 is sufficient to abolish mitophagy almost completely. Therefore, the effects seen on mitophagy following EMC3 knockout in Figure 7 might be smaller in a line that is responsive to NIX mitophagy. It would be beneficial to analyse basal mitophagy flux in an additional cell line, for example U2OS (Fig S1E) in which NIX is responsive to hypoxia.
2. Following on from comment 1 above, Figure 7 would benefit with an analysis of hypoxia (or DFP, or cobalt chloride) stimulation of mitophagy to assess whether mitophagy levels are higher in EMC3 KOs. The authors argue that BNIP3 is trafficked to the ER during mitophagy and is not turned over by mitophagy itself, it would therefore be interesting to test if BNIP3 is prevented from being removed from mitochondria whether this would affect the rate or levels of mitophagy under stimulating conditions.
3. Continuing from comment 2, given that the authors conclude that BNIP3 is not turned over by mitophagy, can they examine whether BNIP3 is excluded from sealed mitophagosomes?
4. Figure 4B: The localisation of tf-BNIP3 is reminiscent of ER in BTZ treated samples. How much of the protein is on mitochondria in the presence of BTZ? Does MLN4924 cause a similar issue?
5. Is the BNIP3(FisTMD) expressed to equivalent levels to WT BFP-BNIP3? Given that the Fis1 form of BNIP3 cannot traffic to endolysosomes, its levels might be higher. In addition, overexpression of the BNIP3-Fis construct was used to make the argument that dimerization is not important for mitophagy. But the authors should also take into account the possibility that

with overexpression, the potential efficiency afforded to mitophagy via dimerization of endogenous proteins may be negated, and therefore hidden. Given this, I don't think that the authors can confidently conclude that dimerization does not contribute to mitophagy, and that instead its main role is ER-endolysosomal turnover of BNIP3.

6. Can the authors assess whether BNIP3 that is on mitochondria is transferred to the ER (perhaps through photoswitchable GFP-BNIP, activated on mitos and then observe its transfer to ER)? This seems important in order to address the possibility that BNIP3 that is being turned over by the endolysosome is being delivered directly to the ER.

****Minor comments:****

1. Figure 3B: Are the red puncta observed in USO1 and SAR1A cells a product of higher levels of ER-phagy owing to BNIP3's high presence on the ER membrane?
2. Please include molecular weight markers for all western blots.
3. Figure 5A-G: These data do not make a convincing case for the role of dimerization and are very difficult to follow. Only the mislocalized S172A mutant was responsive to Baf treatment, while the LG swap mutant which is mitochondrial and cannot dimerize is unaffected by Baf treatment. Figure 5H-I utilise a construct of BNIP3 that is missing most of the protein and which has very low turnover (Figure 5B). Unfortunately these results don't make a highly convincing case about the biology of native, full length, mitochondrial BNIP3. The authors are advised to either strengthen the dimerization argument, or perhaps lighten the language around the main conclusions from these data.

2. Significance:

Significance (Required)

Overall, this is a valuable and important study that provides an important new advance into how mitophagy is regulated by mitophagy receptors. It adds another layer of regulation in addition to the ubiquitin-proteasome mediated restriction of mitophagy reported by others. The data are predominantly convincing and make a strong argument for endolysosomal turnover of BNIP3 to regulate its levels. This study will be of high interest to the field of mitophagy. There is also general interest to the field of mitochondrial biology that a TA mitochondrial (and peroxisomal) protein can be extracted from mitochondria, transferred to the ER, and eventually to the endolysosomal system.

Reviewer expertise: mitophagy mechanisms, autophagosome formation

3. How much time do you estimate the authors will need to complete the suggested revisions:

Estimated time to Complete Revisions (Required)

(Decision Recommendation)

Between 3 and 6 months

Yes

Review #2

1. Evidence, reproducibility and clarity:

Evidence, reproducibility and clarity (Required)

Mitochondria are degraded by autophagy in a reaction that depends on mitophagy receptors, such as BNIP3 and BNIP3L/NIX. These receptors are regulated on the level of gene expression, but also on a post-translational level. This study elucidates the processes which control the levels of BNIP3 in a process that relies on the alternative distribution of the protein to different cellular compartments. BNIP3 is a tail-anchored protein that is inserted into the mitochondrial outer membrane and, alternatively, into the ER membrane. The authors show that insertion into the ER is dependent on the EMC complex and, to a lesser extent, on the GET complex. After insertion into the ER membrane, BNIP3 can be trafficked to lysosomes for degradation or, alternatively, be degraded by proteasomal proteolysis. This study provided evidence that the conditional distribution of BNIP3 to these different intracellular locations is used to control mitophagy even though the specific conditions which determine the alternative destinations remain largely unexplored. The study is of high technical quality, all important controls are shown, and the text is well written.

****Specific points****

1. The mechanism of the alternative distribution is not addressed here. Is the location of BNIP3 dependent on where the newly synthesized protein is initially targeted to (such as in the case of Pink1) or is there a constant redistribution and flux of the protein between the two membranes?

This is an important aspect which should be experimentally addressed and some data to this should be already published as part of this study since this aspect is important for the final model proposed.

2. How is BNIP3 inserted into the outer membrane? A previous study from the Weissman lab proposed that MTCH2 serves as insertase. The authors did not mention MTCH1 and MTCH2 in context of Fig. 2B. Were these proteins not found? Did the authors test the relevance of MTCH2 in their assay? This aspect should be addressed and mentioned.
3. The authors show that BNIP3 on the ER is not stable but degraded by the proteasome. Does this require ERAD factors? Is the mitochondrial BNIP3 protein likewise degraded by proteasomal degradation? It is not clear whether both BNIP3 pools are constantly turned over or whether degradation exclusively/predominantly occurs on the ER surface.
4. The authors generated an interesting BNIP3 mutant with a C-terminal Fis1 anchor. This variant is constantly located in the outer membrane (which is shown here). The physiological consequence of the constitutive distribution on mitochondria is however only superficially studied. The authors should characterize this interesting mutant in some more depth.
5. The results of the screen shown in Fig. 2B are particularly interesting for readers. The glutathione peroxidase GPX4 was found as a top hit among the EMC components. GPX4 protects membranes (including those of mitochondria) against oxidative damage, is a major component of ferroptosis and linked to mitochondrial dysfunction and mitophagy. The authors should mention this interesting hit in the context of their discussion of the lipid-sensing properties of the dimerizing TM domains of BNIP3.

2. Significance:

Significance (Required)

Many studies in the last years focused on the roles of Pink and Parkin in the context of mitophagy, a system that also relies on alternative protein targeting (in that case between the inner and outer membrane of mitochondria). The study here shows that BNIP3, another highly important mitophagy receptor, uses in principle a similar strategy, however, here the alternative targeting occurs between the mitochondrial outer membrane and the ER membrane. Mechanistic insights are provided, for example also into the different domains of BNIP3 and their relevance for targeting and mitophagy. The study therefore addresses an important aspect, is of excellent quality and will be of interest for a broad readership.

3. How much time do you estimate the authors will need to complete the suggested revisions:

Estimated time to Complete Revisions (Required)

(Decision Recommendation)

Between 1 and 3 months

4. Review Commons values the work of reviewers and encourages them to get credit for their work. Select 'Yes' below to register your reviewing activity at Web of Science Reviewer Recognition Service (formerly Publons); note that the content of your review will not be visible on Web of Science.

Yes

Review #3

1. Evidence, reproducibility and clarity:

Evidence, reproducibility and clarity (Required)

In this manuscript, the authors provided a comprehensive study on the regulation of BNIP3 protein levels by both the non-autophagic lysosomal and proteasomal degradation pathways. Using various nifty tools, the authors demonstrated that ER-localised BNIP3 dimers are rerouted via the ER-lysosomal pathway for non-autophagic lysosomal degradation whereas BNIP3 monomers on both the ER and mitochondria are targeted for proteasomal degradation. Together, these pathways help to repress hyperactivation of basal mitophagy.

Overall, I find this project very well executed and the manuscript is also very clear and concise. The key conclusions of this work are supported by orthogonal approaches, thus making the data highly convincing. I only have a few major and several minor comments for this manuscript:

****Major comments:****

1. For all of the tf-BNIP3 FACS data (all violin plots), it is unclear how many biological replicates were performed. The author only stated that at least 10,000 cells were analysed per sample but I believe this is for each biological replicate. To better demonstrate the biological replicates, the authors should consider using bar graphs of the medians (triplicates) with error bars.
2. In Fig 3D, it is unclear as to why there is no basal state accumulation of BNIP3 protein levels compared to Baf1A treated condition especially with USO1 and SAR1A KO samples. Is this because BNIP3 are targeted for proteasomal degradation? I think Fig 3D should include a BTZ treatment next to Baf1A to account for the lack of basal state accumulation of BNIP3.
3. Truncation of proteins could affect their protein stability even during their synthesis. For Fig 5B and 6B, the authors should show the blots for the expression of the different truncated

mutants to prove that the change in BNIP3 stability and their effect of mitoflux (or lack thereof), is not due to poor expression of these mutants.

4. For the data in Fig 7, the authors demonstrated that treating cells with proteasomal inhibitor increases mitoflux. Since the proteasome targets monomeric BNIP3 for degradation, the logical assumption is that BTZ drives dimerization of BNIP3. Can the authors demonstrate this in an approach similar to Fig 5C? This simple experiment will add significant insight into the study.

****Minor comments:****

1. In line 168-169, "In addition, multiple suppressor genes identified from our screen had previously been reported including TMEM11..." -- Unclear what biology they are reported to be involved in

2. Along the line with Major comment 2, the explanation for Fig 3D needs to be better elaborated, perhaps to include the role of proteasome already at this point (if the authors think this is the reason why basal BNIP3 levels remains low with USO1 and SAR1A KO).

3. Line 302-304, I believe that statement only refers to Fig S4C and the statement for Fig 5G is in the next sentence. Please remove Fig5G from line 304. It was confusing to read.

4. Line 367, there is a reference for Fig S5C but that figure is missing.

5. Line 410-411, are there any reported clinical cases of EMC mutations with phenotypes that could be explained by elevated mitophagy?

2. Significance:

Significance (Required)

My expertise lies in organelle-selective autophagy and protein homeostasis. Overall, I think this is a very strong manuscript and the data are very solid. The work adds to our current understanding of the basal regulation of BNIP3 which was not previously explored. The novelty of this work lies in the unexpected regulation of BNIP3 via an autophagy-independent, lysosomal pathway and the observation has the potential to be extended to the regulation of the stability of other tail-anchored proteins. This is a very specialised study and will be of interest to the mitophagy and transmembrane protein regulation community.

3. How much time do you estimate the authors will need to complete the suggested revisions:

Estimated time to Complete Revisions (Required)

(Decision Recommendation)

Less than 1 month

Yes

Revision Plan

Manuscript number: RC-2023-01934

Corresponding author(s): Christopher Shoemaker

1. General Statements

Response: Thank you to all the reviewers for their helpful efforts on behalf of our manuscript. At current, we have addressed most of the reviewers' major comments, including providing additional replicates for many experiments and clarifying ambiguous points in the text. Related data, figures and text have been adjusted accordingly. We believe that these changes have improved our manuscript, both strengthening our main conclusions and clarifying ambiguous text.

Several still-ongoing experiments are elaborated below. These experiments are well within the abilities of our lab and can be completed in short order.

Specific responses to the individual concerns addressed by the reviewers are outlined below.

Please feel free to contact me if I can be of any help in the decision process.

2. Description of the planned revisions

Insert here a point-by-point reply that explains what revisions, additional experimentations and analyses are planned to address the points raised by the referees.

[Reviewer 1]

Comment: Across the manuscript, NIX levels appear to be unresponsive to most treatments in the MDA-MB-231 line, including hypoxia treatment. This is an unusual result and raises questions about the role of NIX in MDA-MB-231 line, mainly that BNIP3 is the primary driver of mitophagy in this system. Indeed, Figure 7D indicates that there is very little mitophagy contribution by NIX since knockout of BNIP3 is sufficient to abolish mitophagy almost completely. Therefore, the effects seen on mitophagy following EMC3 knockout in Figure 7 might be smaller in a line that is responsive to NIX mitophagy. It would be beneficial to analyse basal mitophagy flux in an additional cell line, for example U2OS (FigS1E) in which NIX is responsive to hypoxia.

Response: Thank you for bringing this intriguing insight to our attention. We have seen that EMC3 knockout prevents lysosomal delivery of BNIP3 in U2OS cells (**Fig S2D**). However, we don't know what the effects on mitophagy are in U2OS, or the extent to which mitophagy is dependent on BNIP3 and/or NIX. To test this, we will perform the suggested experiment, taking mt-Keima expressing U2OS cells testing the role of NIX and/or BNIP3 in mitophagy.

Comment: Following on from comment 1 above, Figure 7 would benefit with an analysis of hypoxia (or DFP, or cobalt chloride) stimulation of mitophagy to assess whether mitophagy levels are higher in EMC3 KOs. The authors argue that BNIP3 is trafficked to the ER during mitophagy and is not turned over by mitophagy itself, it would therefore be interesting to test if BNIP3 is

Revision Plan

prevented from being removed from mitochondria whether this would affect the rate or levels of mitophagy under stimulating conditions.

Response: To address this question, we will perform mitoflux analysis on EMC3 KO cells +/- hypoxia.

Comment: Figure 4B: The localisation of tf-BNIP3 is reminiscent of ER in BTZ treated samples. How much of the protein is on mitochondria in the presence of BTZ? Does MLN4924 cause a similar issue?

Response: To address this question, we will perform fluorescence microscopy of tf-BNIP3 cells co-expressing mito-BFP under these treatments and utilize our Coloc2 plugin pipeline to monitor correlation.

Comment: Can the authors assess whether BNIP3 that is on mitochondria is transferred to the ER (perhaps through photoswitchable GFP-BNIP, activated on mitos and then observe its transfer to ER)? This seems important in order to address the possibility that BNIP3 that is being turned over by the endolysosome is being delivered directly to the ER.

Response: This is an interesting question and a curiosity also shared by Reviewer #2. To test this hypothesis, we will utilize a photo-switchable Dendra2 fluorophore to track BNIP3 in the cell via microscopy.

[Reviewer #2]

Comment: How is BNIP3 inserted into the outer membrane? A previous study from the Weissman lab proposed that MTCH2 serves as insertase. The authors did not mention MTCH1 and MTCH2 in context of Fig. 2B. Were these proteins not found? Did the authors test the relevance of MTCH2 in their assay? This aspect should be addressed and mentioned.

Response: Thank you for the insight and suggestion. We were intrigued when the Weissman/Voorhees paper characterizing MTCH1/2 was published. Consistent with their findings, MTCH2 was found in the “suppressor” population of our tf-BNIP3 CRISPR screen, but given our 0.5-fold change threshold, the gene was not validated (fold change value = 0.46, Table S1). We suspect the lack of significance stems from the redundancy with MTCH1. Consequently, we would hypothesize that MTCH1/2 are the responsible insertases. To formally address this suggestion, we plan to genetically perturb MTCH1/2 and look at BNIP3 localization and mitophagy.

Comment: The authors generated an interesting BNIP3 mutant with a C-terminal Fis1 anchor. This variant is constantly located in the outer membrane (which is shown here). The physiological consequence of the constitutive distribution on mitochondria is however only superficially studied. The authors should characterize this interesting mutant in some more depth.

Response: In the original manuscript, we characterized BNIP3(Fis1^{TMD}) for lysosomal delivery and mitophagy. Going forward, we will perform Seahorse oxygen consumption experiments and

Revision Plan

mitochondrial network analysis to view the physiological consequences of constitutive expression of BNIP3(Fis1^{TMD}) on the outer membrane.

3. Description of the revisions that have already been incorporated in the transferred manuscript

Please insert a point-by-point reply describing the revisions that were already carried out and included in the transferred manuscript. If no revisions have been carried out yet, please leave this section empty.

[Reviewer #1]

Comment: Continuing from comment 2, given that the authors conclude that BNIP3 is not turned over by mitophagy, can they examine whether BNIP3 is excluded from sealed mitophagosomes?

Response: We have softened the wording of our conclusions to reflect that the vast majority of BNIP3 lysosomal degradation is by this alternative pathway and not mitophagy. However, we do not wish to completely dismiss that BNIP3 is present on mitophagosomes. Rather, if mitophagosomes contain BNIP3, they seemingly account for only a very small portion of BNIP3 degradation in the cell, to the extent that it is not easily detectable by our assays (Lines 414-419). Definitely identifying whether BNIP3 is in sealed mitophagosomes will be part of future studies using CLEM or FIB-SEM techniques.

Comment: Is the BNIP3(FisTMD) expressed to equivalent levels to WT BFP-BNIP3? Given that the Fis1 form of BNIP3 cannot traffic to endolysosomes, its levels might be higher. In addition, overexpression of the BNIP3-Fis construct was used to make the argument that dimerization is not important for mitophagy. But the authors should also take into account the possibility that with overexpression, the potential efficiency afforded to mitophagy via dimerization of endogenous proteins may be negated, and therefore hidden. Given this, I don't think that the authors can confidently conclude that dimerization does not contribute to mitophagy, and that instead its main role is ER-endolysosomal turnover of BNIP3.

Response: We thank the reviewer for pointing out the possible over-interpretation of our data. Overexpression is an important caveat to consider. We would expect the Fis1 form of BNIP3 to be higher in protein levels given its deficiency in endolysosomal trafficking. Still, as the reviewer points out, over-expression could be mitigating the effect of our dimerization mutants. This caveat is now discussed in the manuscript and our interpretations regarding this fact have been greatly softened (Lines 373-376, Lines 449-462).

Comment: Please include molecular weight markers for all western blots.

Response: All western blots have now been labeled with molecular weight markers.

Comment: Figure 5A-G: These data do not make a convincing case for the role of dimerization and are very difficult to follow. Only the mislocalized S172A mutant was responsive to Baf treatment, while the LG swap mutant which is mitochondrial and cannot dimerize is unaffected by

Revision Plan

Baf treatment. Figure 5H-I utilize a construct of BNIP3 that is missing most of the protein and which has very low turnover (Figure 5B). Unfortunately these results don't make a highly convincing case about the biology of native, full length, mitochondrial BNIP3. The authors are advised to either strengthen the dimerization argument, or perhaps lighten the language around the main conclusions from these data.

Response: Thank you for bringing the lack of clarity to our attention. Both dimer mutants of BNIP3 (S172A and LG swap) are insensitive to Baf-A1 treatment. These results hold for full-length BNIP3 using either the tf (**Fig 5D**) or IRES (**Fig 5I**) reporter. To demonstrate that defects in lysosomal transport were due to dimerization defects (and not other, unanticipated effects of the mutations), we looked at whether chemically induced dimerization could reverse the trafficking defects. Indeed, forced dimerization of the ER-restricted variant rescued ER-to-lysosome trafficking. From this, we conclude that that dimerization is a critical facet of BNIP3 trafficking to the lysosome.

We have re-worked the relevant text (both in results and discussion) to clarify major points and lighten the language around the conclusions from these data (described below).

First, as mentioned above, we have added a significant discussion about the limitations of our assay and of possible interpretations. (Lines 300-303, Lines 323-326, Lines 483-489).

Second, with regards to the specific construct used in this experiment, we have expanded the results section to better describe our rationale and approach (Lines 304-308). In short, because dimerization of native BNIP3 occurs within the membrane, we aimed to place the DmrB domain as close to the TM segment as possible. Due to the topology of TA proteins, a C-terminal tag isn't possible. Therefore, we used the shortest truncation version of BNIP3 (117-end) that undergoes measurable lysosomal delivery. This was an important experimental consideration, and one we did not sufficiently rationalize in the original manuscript. We now include this point in the text.

[Reviewer #2]

Comment: The authors show that BNIP3 on the ER is not stable but degraded by the proteasome. Does this require ERAD factors? Is the mitochondrial BNIP3 protein likewise degraded by proteasomal degradation? It is not clear whether both BNIP3 pools are constantly turned over or whether degradation exclusively/predominantly occurs on the ER surface.

Response: These are fascinating mechanistic questions. We hope to thoroughly address these questions in a subsequent study. However, as a teaser, we have included the basic answer to these questions in **Fig 5I**.

To preliminarily characterize the proteasomal degradation of ER- and mitochondrial-BNIP3, we utilized our IRES reporter system - adapted from Steve Elledge's system for degron monitoring (Fig 5I). Strikingly, our ER-restricted BNIP3 mutation (S172A) is sensitive to inhibition of both the proteasome and the AAA-ATPase p97/VCP, a key extractase for ERAD substrates. These data tentatively suggest an ERAD-dependent degradation mechanism (although many follow-up studies will be needed to confirm the mechanistic details). In sharp contrast, our mitochondrial-restricted mutant (LG Swap) is sensitive to proteasome inhibition by Bortezomib, but it is insensitive to VCP inhibition. The differential requirement for VCP suggests that

Revision Plan

proteasomal degradation occurs on both cellular pools of BNIP3 albeit through different mechanisms.

Comment: The results of the screen shown in Fig. 2B are particularly interesting for readers. The glutathione peroxidase GPX4 was found as a top hit among the EMC components. GPX4 protects membranes (including those of mitochondria) against oxidative damage, is a major component of ferroptosis and linked to mitochondrial dysfunction and mitophagy. The authors should mention this interesting hit in the context of their discussion of the lipid-sensing properties of the dimerizing TM domains of BNIP3.

Response: Thank you to Reviewer #2 for bringing this to our attention. The relationship between GPX4 and BNIP3 flux is very interesting. We have incorporated GPX4 into the discussion section (Lines 457-459).

[Reviewer #3]

Comment: For all of the tf-BNIP3 FACS data (all violin plots), it is unclear how many biological replicates were performed. The author only stated that at least 10,000 cells were analyzed per sample, but I believe this is for each biological replicate. To better demonstrate the biological replicates, the authors should consider using bar graphs of the medians(triplicates) with error bars.

Response: We have included biological replicates of FACS data in all primary figures (except for Fig.1C). Biological replicates, represented as medians (in triplicate), are indicated in figure legends.

Comment: In Fig 3D, it is unclear as to why there is no basal state accumulation of BNIP3 protein levels compared to Baf1A treated condition especially with USO1 and SAR1A KO samples. Is this because BNIP3 are targeted for proteasomal degradation? I think Fig 3D should include a BTZ treatment next to Baf1A to account for the lack of basal state accumulation of BNIP3.

Response: We apologize for the lack of clarity on this point. Yes, the reviewer's interpretation of the data is correct. This point is more clearly elaborated in the text of our revised manuscript (Lines 219-223). Our results indicate that when lysosomal degradation is diminished, the expected increase in total BNIP3 protein levels is attenuated by proteasomal degradation (as evidenced by the hyperstability of BNIP3 upon Bortezomib treatment in mutant backgrounds). As requested, we have included the same knockout panel, now treated with BTZ (**Fig S2E**). These genetic data are further supported by Fig 3E, where a small molecule inhibitor of vesicle trafficking, Brefeldin-A, ameliorates the effect of lysosomal inhibition (BafA1) but exacerbates the effect of proteasome inhibition.

Comment: Truncation of proteins could affect their protein stability even during their synthesis. For Fig 5B and 6B, the authors should show the blots for the expression of the different truncated mutants to prove that the change in BNIP3 stability and their effect of mitoflux (or lack thereof), is not due to poor expression of these mutants.

Response: These were important potential caveats to document, and we thank the reviewer for their comment.

Revision Plan

We note that, due to differences in transduction efficiency, western blot data is an incomplete measure for relative expression levels – it cannot distinguish between fraction of cells transduced and expression level per cell. However, RFP fluorescence (Fig 5B) and BFP fluorescence (Fig 6B) are fluorescent internal controls allowing us to assess expression levels with single cell resolution. We have provided histograms of RFP and/or BFP intensity (**new Fig S4A, Fig S5B**), which provides support that overall expression levels of these constructs are similar. Critically, any variation we observe does not correlate with any of the effects we report.

In addition, we have clarified the figure axis in **Fig 5B** to indicate that the value we are reporting is the “fold-stabilization upon BafA1 treatment”. The original figure legend wasn’t clear. Our metric (fold-stabilization) is internally normalized to compensate for differences in expression level. This is an important clarification.

Comment: For the data in Fig 7, the authors demonstrated that treating cells with proteasomal inhibitor increases mitoflux. Since the proteasome targets monomeric BNIP3 for degradation, the logical assumption is that BTZ drives dimerization of BNIP3. Can the authors demonstrate this in an approach similar to Fig 5C? This simple experiment will add significant insight into the study.

Response: Thank you for the suggestion. As Fig 5C relied on BNIP3 over-expression, we thought it even more informative to assess the effects of BTZ on dimerization of endogenous BNIP3. Indeed, we see accumulation of an SDS-resistant BNIP3 dimer in cells treated with BTZ (**new Fig S2E, line 221**). We hypothesize that BTZ indirectly drives dimerization of BNIP3 by accumulating the total levels of the protein, potentiating monomers to form additional stable dimers.

Comment: In line 168-169, "In addition, multiple suppressor genes identified from our screen had previously been reported including TMEM11..." -- Unclear what biology they are reported to be involved in

Response: We have clarified this line to read: "In addition, we recovered multiple known suppressors of BNIP3 flux, including outer membrane protein spatial restrictor TMEM11, mitochondrial protein import factors DNAJA3 and DNAJA11, and mitochondrial chaperone HSPA9"

Comment: Along the line with Major comment 2, the explanation for Fig 3D needs to be better elaborated, perhaps to include the role of proteasome already at this point (if the authors think this is the reason why basal BNIP3 levels remains low with USO1 and SAR1A KO).

Response: We have included a discussion about compensation by the proteasome in these genetic backgrounds (lines 219-226) and have referred to the newly incorporated western blot (**new Fig S2E**).

Comment: Line 302-304, I believe that statement only refers to Fig S4C and the statement for Fig5G is in the next sentence. Please remove Fig5G from line 304. It was confusing to read.

Response: The reference of **Fig 5G** has been removed.

Revision Plan

Comment: Line 367, there is a reference for Fig S5C but that figure is missing.

Response: The spurious reference has been removed.

Comment: Line 410-411, are there any reported clinical cases of EMC mutations with phenotypes that could be explained by elevated mitophagy?

Response: Thank you for the suggestion. There are clinical presentations of EMC mutations and splice variants in diseases and conditions related to the central nervous system (PMID: 23105016, PMID: 26942288, PMID: 29271071). However, all characterization has been done in the clinical setting looking at clinical presentations/symptoms and not molecular or cellular characterization. We have added a line to the discussion about this speculative correlation between EMC deficiency and mitophagy (lines 516-519).

4. Description of analyses that authors prefer not to carry out

Please include a point-by-point response explaining why some of the requested data or additional analyses might not be necessary or cannot be provided within the scope of a revision. This can be due to time or resource limitations or in case of disagreement about the necessity of such additional data given the scope of the study. Please leave empty if not applicable.

[Reviewer #1]

Comment: Figure 3B: Are the red puncta observed in USO1 and SAR1A cells a product of higher levels of ER-phagy owing to BNIP3's high presence on the ER membrane?

Response: This is an intriguing hypothesis. We will test whether this is true using a USO1/ATG9A dual KO. However, we don't think this result is critical to the overall arc of the manuscript and we will not include these data if they indicate otherwise.

Dear Dr Shoemaker,

Thank you for submitting your work for consideration by the EMBO Journal and transferring your manuscript from Review Commons, now listed as EMBOJ-2023-114702. We have now carefully assessed your manuscript together with the referee reports and your point-by-point response to their concerns. I am happy to say that we find the results to be of interest for the EMBO Journal, and thus are positive to have a revised study re-evaluated by the referees.

Given the experts' overall positive recommendations and based on your detailed response, I am thus pleased to invite you to submit a revised version of the manuscript, addressing the issues raised. We however like to stress that i.p. addressing the concerns of referee #1 seems critical to progress towards publication of this work at the EMBO Journal. I should also add that it is our venue's policy to allow only a single round of revision, and acceptance of your manuscript will therefore depend on the completeness of your responses in this revised version.

When submitting your revised manuscript, please carefully review the instructions below.

Please feel free to approach me any time should you have additional questions related to this.

Thank you for the opportunity to consider your work for publication.

I look forward to your revision.

Kind regards,

Daniel Klimmeck

Daniel Klimmeck, PhD
Senior Editor
The EMBO Journal

Instruction for the preparation of your revised manuscript:

2) individual production quality figure files as .eps, .tif, .jpg (one file per figure).

3) a .docx formatted letter INCLUDING the reviewers' reports and your detailed point-by-point response to their comments. As part of the EMBO Press transparent editorial process, the point-by-point response is part of the Review Process File (RPF), which will be published alongside your paper.

4) a complete author checklist, which you can download from our author guidelines ([https://wol-prod-cdn.literatumonline.com/pb-assets/embo-site/Author Checklist%20-%20EMBO%20J-1561436015657.xlsx](https://wol-prod-cdn.literatumonline.com/pb-assets/embo-site/Author%20Checklist%20-%20EMBO%20J-1561436015657.xlsx)). Please insert information in the checklist that is also reflected in the manuscript. The completed author checklist will also be part of the RPF.

6) It is mandatory to include a 'Data Availability' section after the Materials and Methods. Before submitting your revision, primary datasets produced in this study need to be deposited in an appropriate public database, and the accession numbers and database listed under 'Data Availability'. Please remember to provide a reviewer password if the datasets are not yet public (see <https://www.embopress.org/page/journal/14602075/authorguide#datadeposition>).

7) Our journal encourages inclusion of *data citations in the reference list* to directly cite datasets that were re-used and obtained from public databases. Data citations in the article text are distinct from normal bibliographical citations and should directly link to the database records from which the data can be accessed. In the main text, data citations are formatted as follows: "Data ref: Smith et al, 2001" or "Data ref: NCBI Sequence Read Archive PRJNA342805, 2017". In the Reference list, data citations must be labeled with "[DATASET]". A data reference must provide the database name, accession number/identifiers and a resolvable link to the landing page from which the data can be accessed at the end of the reference. Further instructions are available at .

8) At EMBO Press we ask authors to provide source data for the main and EV figures. Our source data coordinator will contact you to discuss which figure panels we would need source data for and will also provide you with helpful tips on how to upload and organize the files.

Numerical data can be provided as individual .xls or .csv files (including a tab describing the data). For 'blots' or microscopy, uncropped images should be submitted (using a zip archive or a single pdf per main figure if multiple images need to be supplied for one panel). Additional information on source data and instruction on how to label the files are available at .

9) We replaced Supplementary Information with Expanded View (EV) Figures and Tables that are collapsible/expandable online (see examples in <https://www.embopress.org/doi/10.15252/emboj.201695874>). A maximum of 5 EV Figures can be typeset. EV Figures should be cited as 'Figure EV1, Figure EV2' etc. in the text and their respective legends should be included in the main text after the legends of regular figures.

11) For data quantification: please specify the name of the statistical test used to generate error bars and P values, the number (n) of independent experiments (specify technical or biological replicates) underlying each data point and the test used to calculate p-values in each figure legend. The figure legends should contain a basic description of n, P and the test applied. Graphs must include a description of the bars and the error bars (s.d., s.e.m.).

We realize that it is difficult to revise to a specific deadline. In the interest of protecting the conceptual advance provided by the work, we recommend a revision within 3 months (15th Sep 2023). Please discuss the revision progress ahead of this time with the editor if you require more time to complete the revisions.

Manuscript number: EMBOJ-2023-114702

Corresponding author(s): Christopher Shoemaker

Dear Daniel,

We are pleased to present the enclosed revised research manuscript entitled “The EMC governs lysosomal turnover of a mitochondrial TA protein, BNIP3, to restrict mitophagy.”

Let me begin by thanking you and the reviewers for your efforts on behalf of our manuscript. Following reviewer advice, we have made many improvements to our manuscript. Overall, we have revised the manuscript thoroughly. Related data, figures and text have been adjusted accordingly, with major changes indicated (blue) directly in the text. Notably, the outcomes of the proposed experiments added to, but did not appreciably change, the main findings or conclusions of this manuscript. We are confident that the changes made in response to the reviewers' feedback have significantly elevated the rigor and impact of our work. We hope that our manuscript has improved and will meet the high standards of EMBO J.

Specific responses to the individual concerns addressed by the reviewers are outlined below. Please feel free to contact me if I can be of any help in the decision process.

Sincerely,

Chris Shoemaker

[Reviewer 1]

Major Comments:

Comment 1: Across the manuscript, NIX levels appear to be unresponsive to most treatments in the MDA-MB-231 line, including hypoxia treatment. This is an unusual result and raises questions about the role of NIX in MDA-MB-231 line, mainly that BNIP3 is the primary driver of mitophagy in this system. Indeed, Figure 7D indicates that there is very little mitophagy contribution by NIX since knockout of BNIP3 is sufficient to abolish mitophagy almost completely. Therefore, the effects seen on mitophagy following EMC3 knockout in Figure 7 might be smaller in a line that is responsive to NIX mitophagy. It would be beneficial to analyse basal mitophagy flux in an additional cell line, for example U2OS (Fig EV1E) in which NIX is responsive to hypoxia.

Response: This is an interesting insight. As the reviewer notes, we observed very little effect on NIX in MDA-MB-231 cells and a more substantial effect on NIX in U2OS cells (Fig EV2F). To see how this might affect mitophagy differentially in these cell lines, we performed the suggested experiment by taking mt-Keima expressing U2OS cells and testing the role of NIX and/or BNIP3 in both basal and induced mitophagy. In contrast to MDA-MB-231 cells, we found that genetic perturbation of either BNIP3 or NIX in U2OS cells diminished both basal and BTZ-induced mitophagy (new Fig EV6D). These findings align with the existing literature, where basal mitophagy in U2OS cells is reported to respond additively to BNIP3 and NIX. Further investigation will be required to identify the molecular signals that explain cell-type specific differences in basal mitophagy. These results and the relevant references are now discussed in lines 405-408 of the manuscript.

Comment 2: Following on from comment 1 above, Figure 7 would benefit with an analysis of hypoxia (or DFP, or cobalt chloride) stimulation of mitophagy to assess whether mitophagy levels are higher in EMC3 KOs. The authors argue that BNIP3 is trafficked to the ER during mitophagy and is not turned over by mitophagy itself, it would therefore be interesting to test if BNIP3 is prevented from being removed from mitochondria whether this would affect the rate or levels of mitophagy under stimulating conditions.

Response: As proposed, we performed mitoflux analysis on EMC3 KO cells +/- hypoxia (1% O₂, 18h). Consistent with BTZ treatment, hypoxia treatment of EMC3 KO cells resulted in an additive increase in mitophagy as compared to controls (new Fig EV6F). These data are consistent with the interpretation that, under mitophagy stimulating conditions (e.g. either proteostatic collapse or hypoxia-induced transcriptional rewiring), ER-insertion of BNIP3 similarly regulates mitophagy levels.

Comment 3: Continuing from comment 2, given that the authors conclude that BNIP3 is not turned over by mitophagy, can they examine whether BNIP3 is excluded from sealed mitophagosomes?

Response: We have softened the wording of our conclusions to reflect that the vast majority of BNIP3 lysosomal degradation is by this alternative pathway and not mitophagy. However, we do not wish to completely dismiss that BNIP3 is present on mitophagosomes. Indeed, our data confirm that the LIR of BNIP3 is required for its mitophagy function. As such, BNIP3 is likely functioning like a traditional autophagy receptor. And yet, we would like to highlight that this fraction of BNIP3 seemingly accounts for only a very small portion of BNIP3 degradation in the cell, to the extent that it is not easily detectable by our assays (Lines 434-437). Definitively identifying whether BNIP3 is in sealed mitophagosomes will be part of future studies using CLEM or FIB-SEM techniques.

Comment 4: Figure 4B: The localisation of tf-BNIP3 is reminiscent of ER in BTZ treated samples. How much of the protein is on mitochondria in the presence of BTZ? Does MLN4924 cause a similar issue?

Response: To address this question, we performed fluorescence microscopy of tf-BNIP3 cells co-expressing mito-BFP under these treatments and utilized our Coloc2 plugin pipeline to monitor correlation. Treatment with BTZ increased the percentage of BNIP3 protein present on the mitochondria (compare Pearson's Coefficient of 0.34 Veh to 0.46 BTZ) (Fig EV3C), even though we also see concurrent accumulation on ER-like structures. In the case of MLN-4924, we see an even stronger increase in BNIP3 co-localization with our mitochondrial marker (compare Pearson's Coefficient of 0.34 Veh to 0.61 MLN4924). We also note that we do not observe a similar ER accumulation in MLN4924 treated cells, suggesting Nedd8 and neddylation may act primarily on the mitochondria-localized BNIP3 population. The proteasomal regulation of BNIP3 at the surface of each organelle is an area of ongoing investigation.

Comment 5: Is the BNIP3(FisTMD) expressed to equivalent levels to WT BFP-BNIP3? Given that the Fis1 form of BNIP3 cannot traffic to endolysosomes, its levels might be higher. In addition, overexpression of the BNIP3-Fis construct was used to make the argument that dimerization is not important for mitophagy. But the authors should also take into account the possibility that with overexpression, the potential efficiency afforded to mitophagy via dimerization of endogenous proteins may be negated, and therefore hidden. Given this, I don't think that the authors can confidently conclude that dimerization does not contribute to mitophagy, and that instead its main role is ER-endolysosomal turnover of BNIP3.

Response: We thank the reviewer for pointing out the possible over-interpretation of our data. Overexpression is an important caveat to consider. We might expect BNIP3-Fis1 to be present at higher levels given its deficiency in endolysosomal trafficking. When looking at expression levels via TagBFP intensity by flow cytometry, we see that the expression of BNIP3-Fis1 is comparable to wild type BNIP3 (new Fig EV5D). Still, as the reviewer points out, over-expression could be mitigating the effect of our dimerization mutants.

This caveat is now discussed in the manuscript and our interpretations regarding this fact have been greatly softened (Lines 374-378, Lines 468-477).

Comment 6: Can the authors assess whether BNIP3 that is on mitochondria is transferred to the ER (perhaps through photoswitchable GFP-BNIP, activated on mitosis and then observe its transfer to ER)? This seems important in order to address the possibility that BNIP3 that is being turned over by the endolysosome is being delivered directly to the ER.

Response: This is an interesting question and a curiosity also shared by Reviewer #2 (comment 1). A shared response to both comments is provided here:

We ourselves have been very curious about the relative contribution of TA protein extraction and exchange (an idea taken from McKenna et al, PMID: 32973005, and others) vs TA protein mis-targeting into the ER.

As a first attempt to address this question, we utilized a photo-switchable Dendra2 fluorophore to track BNIP3 in the cell via microscopy. We tracked converted Dendra2-BNIP3 in MDA-MB-231 cells over 6h, fixed the cells, and immunostained for lysosomal marker LAMP1. However, the results of these experiments were unclear. First, we found that the Dendra2-BNIP3 construct unexpectedly behaved poorly - mitochondrial localization was rare and Dendra2-BNIP3 was often seen in the endolysosomal pathway immediately. Second, the Dendra2 signal was incredibly stable in lysosomes, with its stability far exceeding our 6h time-lapse. We note that this is contrary to a prior report indicating that converted Dendra2 is pH sensitive, which was our rationale for using this variant in the first place (PMID: 28986251).

With these caveats in mind, we used brefeldin A (BFA) to block new delivery of BNIP3 to the lysosome via the endolysosomal pathway. With this control we saw results that could suggest BNIP3 can be extracted from the mitochondria and delivered to lysosomes: the correlation coefficient between converted Dendra2 and LAMP1 dropped as expected in the BFA-treated cells (0.75 vehicle control vs 0.4 BFA). We are happy to make the data available if the reviewers disagree with its potential utility. However, these data did not rise to the quality we expect from our lab, and we do not wish to include this approach in the manuscript due to its inconclusiveness and potential to distract from the main points. Therefore, we can neither confirm nor exclude the possibility of cellular membrane transfer as described previously by others.

Unsatisfied by this, we wished to revisit this question by asking a related question: "Does BNIP3 in the ER need to be targeted to mitochondria first?"

This proved to be a significantly more tractable approach. In short, we find that:

- 1) Genetic ablation of the mitochondrial TA insertases, MTCH1/2, prevents insertion of BNIP3 onto the OMM and redirects BNIP3 to the ER membrane (new Fig EV2E).
- 2) Under these conditions, lysosomal delivery of BNIP3 is not perturbed (new Fig EV2D).

Together, these data indicate that initial mis-insertion into the ER membrane is sufficient to promote endolysosomal trafficking of BNIP3 without a need to be inserted into the OMM first (Fig EV2D-E). We now discuss this in lines 507-511.

Relatedly, we find that an ER-inserted BNIP3 mutant (S172A) can be directed to the lysosome upon chemically induced dimerization (Fig 5D-G). Thus, ER-localized BNIP3 can exit the ER without ever going through a mitochondrial intermediate.

Reflecting the aforementioned findings, we have changed our model (Fig 7) to include dashed lines for possible extraction events. We feel this modification acknowledges the work and models developed by our colleagues, while also acknowledging it as an area of future study.

Minor comments:

Comment 1: Figure 3B: Are the red puncta observed in USO1 and SAR1A cells a product of higher levels of ER-phagy owing to BNIP3's high presence on the ER membrane?

Response: This is an intriguing hypothesis and one we have pursued for a while. We have looked many ways for evidence that BNIP3 accumulation on the ER leads to aberrant ER-phagy, however we have not found clear evidence supporting this hypothesis. This idea will continue to be tested in our future studies on BNIP3 regulation. Fortunately, we don't think this result is critical to the overall arc of the current manuscript.

Comment 2: Please include molecular weight markers for all western blots.

Response: All western blots have now been labeled with molecular weight markers.

Comment 3: Figure 5A-G: These data do not make a convincing case for the role of dimerization and are very difficult to follow. Only the mislocalized S172A mutant was responsive to Baf treatment, while the LG swap mutant which is mitochondrial and cannot dimerize is unaffected by Baf treatment. Figure 5H-I utilize a construct of BNIP3 that is missing most of the protein and which has very low turnover (Figure 5B). Unfortunately, these results don't make a highly convincing case about the biology of native, full length, mitochondrial BNIP3. The authors are advised to either strengthen the dimerization argument, or perhaps lighten the language around the main conclusions from these data.

Response: Thank you for bringing the lack of clarity to our attention. Both dimer mutants of BNIP3 (S172A and LG swap) are insensitive to Baf-A1 treatment. These results hold for full length BNIP3 using both the tf (Fig 5D) or IRES (Fig 5I) reporter. To demonstrate that defects in lysosomal transport were due to dimerization defects (and not other, unanticipated effects of the mutations), we looked at whether chemically induced dimerization could reverse the trafficking defects. Indeed, forced dimerization of the ER-

restricted variant (i.e. S172A) rescued ER-to-lysosome trafficking. From this, we conclude that that dimerization is a critical facet of BNIP3 export from the ER and/or it's subsequent trafficking to the lysosome.

We have re-worked the relevant text (both in results and discussion) to clarify major points and lighten the language around the conclusions from these data (described below).

First, as mentioned above, we have added a significant discussion about the limitations of our assay and of possible interpretations. (Lines 304-308, Lines 330-332, Lines 503-505).

Second, with regards to the specific construct used in this experiment, we have expanded the results section to better describe our rationale and approach (Lines 304-308). In short, because dimerization of native BNIP3 occurs within the membrane, we aimed to place the DmrB domain as close to the TM segment as possible. Due to the topology of TA proteins, a C-terminal tag isn't possible. Therefore, we used the shortest truncated version of BNIP3 (117-end) that undergoes measurable lysosomal delivery. This was an important experimental consideration, and one we did not sufficiently rationalize in the original manuscript. We now include this point in the text.

[Reviewer #2]

Comment 1: The mechanism of the alternative distribution is not addressed here. Is the location of BNIP3 dependent on where the newly synthesized protein is initially targeted to (such as in the case of Pink1) or is there a constant redistribution and flux of the protein between the two membranes? This is an important aspect which should be experimentally addressed and some data to this should be already published as part of this study since this aspect is important for the final model proposed.

Response: This is an interesting question and a curiosity also shared by Reviewer #1 (comment 6). A shared response to both comments is provided here:

We ourselves have been very curious about the relative contribution of TA protein extraction and exchange (an idea taken from McKenna et al, PMID: 32973005, and others) vs TA protein mis-targeting into the ER.

As a first attempt to address this question, we utilized a photo-switchable Dendra2 fluorophore to track BNIP3 in the cell via microscopy. We tracked converted Dendra2-BNIP3 in MDA-MB-231 cells over 6h, fixed the cells, and immunostained for lysosomal marker LAMP1. However, the results of these experiments were unclear. First, we found that the Dendra2-BNIP3 construct unexpectedly behaved poorly - mitochondrial localization was rare and Dendra2-BNIP3 was often seen in the endolysosomal pathway immediately. Second, the Dendra2 signal was incredibly stable in lysosomes, with its stability far exceeding our 6h time-lapse. We note that this is contrary to a prior report indicating that converted Dendra2 is pH sensitive, which was our rationale for using this variant in the first place (PMID: 28986251).

With these caveats in mind, we used brefeldin A (BFA) to block new delivery of BNIP3 to the lysosome via the endolysosomal pathway. With this control we saw results that could suggest BNIP3 can be extracted from the mitochondria and delivered to lysosomes: the correlation coefficient between converted Dendra2 and LAMP1 dropped as expected in the BFA-treated cells (0.75 vehicle control vs 0.4 BFA). We are happy to make the data available if the reviewers disagree with its potential utility. However, these data did not rise to the quality we expect from our lab, and we do not wish to include this approach in the manuscript due to its inconclusiveness and potential to distract from the main points. Therefore, we can neither confirm nor exclude the possibility of cellular membrane transfer as described previously by others.

Unsatisfied by this, we wished to revisit this question by asking a related question: “Does BNIP3 in the ER need to be targeted to mitochondria first?”

This proved to be a significantly more tractable approach. In short, we find that:

- 3) Genetic ablation of the mitochondrial TA insertases, MTCH1/2, prevents insertion of BNIP3 onto the OMM and redirects BNIP3 to the ER membrane (new Fig EV2E).
- 4) Under these conditions, lysosomal delivery of BNIP3 is not perturbed (new Fig EV2D).

Together, these data indicate that initial mis-insertion into the ER membrane is sufficient to promote endolysosomal trafficking of BNIP3 without a need to be inserted into the OMM first (Fig EV2D-E). We now discuss this in lines 509-511.

Relatedly, we find that an ER-inserted BNIP3 mutant (S172A) can be directed to the lysosome upon chemically induced dimerization (Fig 5D-G). Thus, ER-localized BNIP3 can exit the ER without ever going through a mitochondrial intermediate.

Reflecting the aforementioned findings, we have changed our model (Fig 7) to include dashed lines for possible extraction events. We feel this modification acknowledges the work and models developed by our colleagues, while also acknowledging it as an area of future study.

Comment 2: How is BNIP3 inserted into the outer membrane? A previous study from the Weissman lab proposed that MTCH2 serves as insertase. The authors did not mention MTCH1 and MTCH2 in context of Fig. 2B. Were these proteins not found? Did the authors test the relevance of MTCH2 in their assay? This aspect should be addressed and mentioned.

Response: We were intrigued when the Weissman/Voorhees paper characterizing MTCH1/2 was published. Consistent with their findings, MTCH2 was found in the “suppressor” population of our tf-BNIP3 CRISPR screen, but given our 0.5 log fold-change threshold, the gene was not validated in our initial round (fold change value = 0.46, Table EV1). We suspect the lack of significance stems from the redundancy with

MTCH1. Consequently, we would hypothesize that MTCH1/2 are the responsible insertases. To test this, we genetically perturbed MTCH1/2 and looked at BNIP3 localization, trafficking, and mitophagy.

First, using GFP-BNIP3 expressing cells, we looked at localization differences. In cells with single deletions of MTCH1 or MTCH2, BNIP3 remained localized at mitochondria, consistent with redundant functions and/or compensation between MTCH1 and MTCH2. In contrast, concurrent deletion of both MTCH1 and MTCH2 resulted in a shift in BNIP3 localization to a reticular, ER-like localization (Fig EV2E), consistent with the recently published MTCH paper.

Functionally, we observed similar trends when using mitophagy induction as a functional readout for BNIP3 insertion into the mitochondrial membrane. Single deletion of MTCH1 or MTCH2 revealed partial mitophagy defects, while dual deletion of MTCH1/2 completely ablated BNIP3-dependent mitophagy (e.g. in response to BTZ (Fig EV6C)). Therefore, as predicted, when BNIP3 mitochondrial insertion is disrupted it can no longer induce mitophagy.

Finally, using our tf-BNIP3 reporter cells, we observe continued lysosomal trafficking of BNIP3 upon disruption of mitochondrial insertion (Fig EV2D). The implications of this are discussed in detail in our response to comment 1.

In total, we believe this is a nice packet of data that helps build out our model. Consequently, the role of MTCH1/2 in BNIP3 regulation is now included in our model (Fig 7). Thank you for the suggestion.

Comment 3: The authors generated an interesting BNIP3 mutant with a C-terminal Fis1 anchor. This variant is constantly located in the outer membrane (which is shown here). The physiological consequence of the constitutive distribution on mitochondria is however only superficially studied. The authors should characterize this interesting mutant in some more depth.

Response: Thank you for the suggestion. Indeed, in the original manuscript, we characterized BNIP3(Fis1TMD) only for lysosomal delivery (Fig 6C) and mitophagy (Fig 6D), without considering downstream physiological consequences. To address this, we performed Seahorse oxygen consumption experiments to assess the physiological consequences of constitutive expression of BNIP3(Fis1TMD) on the outer membrane. Consistent with our mtKeima data, BNIP3(Fis1TMD) depleted oxygen consumption comparable to the wild-type (Fig EV5E). These data are consistent with our thesis that the conserved TMD of BNIP3 is not critical for the mitophagic activity of BNIP3, but rather the cell's ability to employ the endolysosomal system to regulate BNIP3 localization and function (line 376-378).

Comment 4: The authors show that BNIP3 on the ER is not stable but degraded by the proteasome. Does this require ERAD factors? Is the mitochondrial BNIP3 protein likewise degraded by proteasomal degradation? It is not clear whether both BNIP3 pools are constantly turned over or whether degradation exclusively/predominantly occurs on the ER surface.

Response: These are fascinating mechanistic questions getting at the fundamental principles governing membrane protein stability (particularly at the mitochondrial surface). We hope to thoroughly address these questions in a subsequent study. However, as a teaser, we have included the basic answer to these questions in Fig 5I.

To preliminarily characterize the proteasomal degradation of ER- and mitochondrial-BNIP3, we utilized our IRES reporter system - adapted from Steve Elledge's system for degron monitoring (Fig 5I). Strikingly, our ER-restricted BNIP3 mutation (S172A) is sensitive to inhibition of both the proteasome and the AAA-ATPase p97/VCP, a key extractase for ERAD substrates. These data tentatively suggest an ERAD-M dependent degradation mechanism (although many follow-up studies will be needed to confirm the mechanistic details). In sharp contrast, our mitochondrial-restricted mutant (LG Swap) is sensitive to proteasome inhibition by Bortezomib, but it is insensitive to VCP inhibition. The differential requirement for VCP suggests that proteasomal degradation occurs on both cellular pools of BNIP3 albeit through different mechanisms.

Comment 5: The results of the screen shown in Fig. 2B are particularly interesting for readers. The glutathione peroxidase GPX4 was found as a top hit among the EMC components. GPX4 protects membranes (including those of mitochondria) against oxidative damage, is a major component of ferroptosis and linked to mitochondrial dysfunction and mitophagy. The authors should mention this interesting hit in the context of their discussion of the lipid-sensing properties of the dimerizing TM domains of BNIP3.

Response: Thank you to Reviewer #2 for bringing this to our attention. The relationship between GPX4 and BNIP3 flux is very promising and one we are now interested in following up on long term. We have incorporated GPX4 into the discussion section (Lines 473-477).

[Reviewer #3]

Major Comments:

Comment 1: For all of the tf-BNIP3 FACS data (all violin plots), it is unclear how many biological replicates were performed. The author only stated that at least 10,000 cells were analyzed per sample, but I believe this is for each biological replicate. To better demonstrate the biological replicates, the authors should consider using bar graphs of the medians(triplicates) with error bars.

Response: We have included biological replicates of FACS data in all primary figures (except for Fig.1C). Biological replicates, represented as medians (in triplicate), are

indicated in figure legends. Notably, no interpretations of our data were changed in the process, however these changes significantly improve our story with regards to rigor and reproducibility. We thank the reviewer for this suggestion.

Comment 2: In Fig 3D, it is unclear as to why there is no basal state accumulation of BNIP3 protein levels compared to Baf1A treated condition especially with USO1 and SAR1A KO samples. Is this because BNIP3 are targeted for proteasomal degradation? I think Fig 3D should include a BTZ treatment next to Baf1A to account for the lack of basal state accumulation of BNIP3.

Response: We apologize for the lack of clarity on this point. Yes, the reviewer's interpretation of the data is correct. This point is more clearly elaborated in the text of our revised manuscript (Lines 219-226). Our results indicate that when lysosomal degradation is diminished, the expected increase in total BNIP3 protein levels is attenuated by proteasomal degradation (as evidenced by the hyperstability of BNIP3 upon Bortezomib treatment in mutant backgrounds). As requested, we have included the same knockout panel, now treated with BTZ (Fig EV2G). These genetic data are further supported by Fig 3E, where a small molecule inhibitor of vesicle trafficking, Brefeldin-A, ameliorates the effect of lysosomal inhibition (BafA1) but exacerbates the effect of proteasome inhibition.

Comment 3: Truncation of proteins could affect their protein stability even during their synthesis. For Fig 5B and 6B, the authors should show the blots for the expression of the different truncated mutants to prove that the change in BNIP3 stability and their effect of mitoflux (or lack thereof), is not due to poor expression of these mutants.

Response: These were important potential caveats to document, and we thank the reviewer for their comment.

We note that, due to differences in transduction efficiency, western blot data were an incomplete measure for relative expression levels – it cannot distinguish between fraction of cells transduced and expression level per cell. However, RFP fluorescence (Fig 5B) and BFP fluorescence (Fig 6B) are fluorescent internal controls allowing us to assess expression levels with single cell resolution. We have provided histograms of RFP and/or BFP intensity (new Fig EV4A, Fig EV5B, Fig EV5E), which provides support that overall expression levels of these constructs are similar. Critically, the variation we observe does not correlate with any of the effects we report.

In addition, we have clarified the figure axis in Fig 5B to indicate that the value we are reporting is the “fold-stabilization upon BafA1 treatment”. The original figure legend wasn't clear. Our metric (fold-stabilization) is internally normalized to further compensate for differences in expression level. This is an important clarification.

Comment 4: For the data in Fig 7, the authors demonstrated that treating cells with proteasomal inhibitor increases mitoflux. Since the proteasome targets monomeric BNIP3 for degradation, the logical assumption is that BTZ drives dimerization of BNIP3. Can the authors demonstrate this in an approach similar to Fig 5C? This simple experiment will add significant insight into the study.

Response: Thank you for the suggestion. As Fig 5C relied on BNIP3 over-expression, we thought it even more informative to assess the effects of BTZ on dimerization of endogenous BNIP3. Indeed, we see accumulation of an SDS-resistant BNIP3 dimer in cells treated with BTZ (new Fig EV2G, line 223). We hypothesize that BTZ indirectly drives dimerization of BNIP3 by accumulating total protein levels, potentiating monomers to form additional stable dimers (line 242).

Minor comments

Comment 1: In line 168-169, "In addition, multiple suppressor genes identified from our screen had previously been reported including TMEM11..." -- Unclear what biology they are reported to be involved in

Response: We have clarified this line to read: "In addition, we recovered multiple known suppressors of BNIP3 flux, including mitochondrial protein import factors DNAJA3 and DNAJA11, the mitochondrial chaperone HSPA9 and, as recently reported, the outer membrane protein TMEM11."

Comment 2: Along the line with Major comment 2, the explanation for Fig 3D needs to be better elaborated, perhaps to include the role of proteasome already at this point (if the authors think this is the reason why basal BNIP3 levels remains low with USO1 and SAR1A KO).

Response: We have included a discussion about compensation by the proteasome in these genetic backgrounds (lines 219-221) and have referred to the newly incorporated western blot (new Fig EV2G).

Comment 3: Line 302-304, I believe that statement only refers to Fig EV4C and the statement for Fig5G is in the next sentence. Please remove Fig5G from line 304. It was confusing to read.

Response: The reference of Fig 5G has been removed.

Comment 4: Line 367, there is a reference for Fig EV5C but that figure is missing.

Response: The spurious reference has been removed.

Comment 5: Line 410-411, are there any reported clinical cases of EMC mutations with phenotypes that could be explained by elevated mitophagy?

Response: Thank you for the suggestion. There are clinical presentations of EMC mutations and splice variants in diseases and conditions related to the central nervous system (PMID: 23105016, PMID: 26942288, PMID: 29271071). However, all characterization has been done in the clinical setting looking at clinical presentations/symptoms and not molecular or cellular characterization. We have added a line to the discussion about this speculative correlation between EMC deficiency and mitophagy (lines 535-537).

Dear Dr Shoemaker,

Thank you for submitting your revised manuscript (EMBOJ-2023-114702R) to The EMBO Journal. As mentioned, your amended study was sent back to the referees for their re-evaluation, and we have received comments from two of them, which I enclose below. Please note that while referee #3 was unfortunately not able to re-evaluate the work at this time, we have assessed your response to the concerns raised editorially and found them to be addressed satisfactorily. As you will see, the other experts stated that the work has been substantially improved by the revisions and they are now broadly in favour of publication, pending minor revision.

Thus, we are pleased to inform you that your manuscript has been accepted in principle for publication in The EMBO Journal.

Please consider the remaining discussion points by referee #2 carefully and adjust the text where appropriate.

Also, we now need you to take care of a number of minor issues related to formatting and data presentation as detailed below, which should be addressed at re-submission.

Please contact me at any time if you have additional questions related to below points.

As you might have noted on our web page, every paper at the EMBO Journal now includes a 'Synopsis', displayed on the html and freely accessible to all readers. The synopsis includes a 'model' figure as well as 2-5 one-short-sentence bullet points that summarize the article. I would appreciate if you could provide this figure and the bullet points.

Thank you for giving us the chance to consider your manuscript for The EMBO Journal. I look forward to your final revision.

Again, please contact me at any time if you need any help or have further questions.

with
Best regards,

Daniel Klimmeck

>> Please add a 'Disclosure and Competing Interests Statement' section to your manuscript.

>> Adjust the reference format to EMBO Journal style, limiting to 10 authors et al. .

>> Recheck publication status of bioRxiv entry: Echavarria-Consuegra L et al (2023).

>> EV figures: You can keep up to 5 EV Figures. The figure files need to be uploaded as individual, high-resolution figure files and labeled "Figure EV1" etc., and the legends need to be added to the manuscript, after the main figure legends. Any additional content should be merged with their legends into one 'Appendix' PDF and renamed "Appendix Figure S1" etc. The appendix will need a ToC with page numbers on the first page. The source data files and figure callouts in the manuscript will need to be adjusted accordingly.

>>Callouts: Tables are incorrectly called out as Table S1-S2, and should be Dataset EV1 and Table EV1.

>> Author Checklist: specify cell line authentication status.

>> Source data: please provide additional source data for figure panels 1B, 6A and 7B.

>> Dataset EV legends: Table EV1 should be renamed to Dataset EV1 with the legend inserted as a separate sheet and with

appropriate callout. Table EV2 should be renamed to Table EV1 with the corresponding callout

>> Please consider additional changes and comments from our production team as indicated below:

- Figure Legends (main + EV):

"Please note that the figure legend style does not comply with the journal guidelines i.e. all the figure legends are in a run-on style."

"1. Please indicate the statistical test used for data analysis in the legends of figures 2b; 5a

2. Please note that in figure 3e there is a mismatch between the annotated p values in the figure legend and the annotated p values in the figure file that should be corrected.

3. Please define the annotated p values **/* in the legends of figures 5a; 6c as appropriate."

"1. Please note that information related to n is missing in the legend of figures 2b; 5b; 6e; EV1h; EV5e

2. Please note that the error bars are not defined in the legend of figures 5b; 6e; EV5e"

Revision to The EMBO Journal should be submitted online within 90 days, unless an extension has been requested and approved by the editor; please click on the link below to submit the revision online before 17th Jan 2024:

Link Not Available

Referee #1:

The authors satisfactorily addressed all points raised on the original version of the study. I fully support the publication of this interesting manuscript in EMBO Journal which elucidates the molecular interactions of ER and mitochondrial membranes in protein trafficking. It considerably advances our understanding of the cooperation and competition of different insertion systems for tail-anchored membrane proteins.

Referee #2:

This is a revised manuscript describing the role of the endolysosomal system in regulating mitophagy receptor levels. Overall, the authors have thoroughly addressed the comments, and while open questions still remain, the study represents an important advance in not only our understanding of mitophagy regulation, but also the regulation of mitochondrial tail-anchored proteins. The study is likely to appeal to a broad audience and is clearly suitable for EMBO J.

Given that mysteries still remain regarding how the majority of BNIP3 is not turned over by mitophagy (i.e potentially escaping mitophagosomes to the ER for endolysosomal turnover), it would be worthwhile to include a brief discussion of this open question as an area for future work. The authors may also wish to refer to precedent for this observation for other mitochondrial proteins during mitophagy (PMID: 23361001). For BNIP3, it would need to reside at mitochondria long enough to initiate mitophagy, before being shipped off to the ER for endolysosomal turnover.

Rev_Com_number: RC-2023-01934

New_manu_number: EMBOJ-2023-114702R

Corr_author: Shoemaker

Title: The EMC governs lysosomal turnover of a mitochondrial TA protein, BNIP3, to restrict mitophagy

(EMBOJ-2023-114702R)
Point-by-Point Response

>> Please add a 'Disclosure and Competing Interests Statement' section to your manuscript.

This section has been included after the acknowledgments section. The authors have no conflict of interests to declare.

>> Adjust the reference format to EMBO Journal style, limiting to 10 authors et al. .

The reference format has been changed to the EMBO Journal style throughout the manuscript.

>> Recheck publication status of bioRxiv entry: Echavarria-Consuegra L et al (2023).

The publication status has not changed and remains as a preprint on bioRxiv.

>> EV figures: You can keep up to 5 EV Figures. The figure files need to be uploaded as individual, high-resolution figure files and labeled "Figure EV1" etc., and the legends need to be added to the manuscript, after the main figure legends. Any additional content should be merged with their legends into one 'Appendix' PDF and renamed "Appendix Figure S1" etc. The appendix will need a ToC with page numbers on the first page. The source data files and figure callouts in the manuscript will need to be adjusted accordingly.

Previously labeled as "Figure EV6" has now been changed to Appendix Figure S1 and included in the appendix with a table of contents. Changes have been made throughout the manuscript.

>>Callouts: Tables are incorrectly called out as Table S1-S2, and should be Dataset EV1 and Table EV1.

The file names have been changed.

>> Author Checklist: specify cell line authentication status.

The cell line authentication status of each cell line used in this study has been clarified in the Tissue Culture subsection of the Materials and Methods.

>> Source data: please provide additional source data for figure panels 1B, 6A and 7B.

We have published all our source data for flow cytometry, microscopy images, and fastq sequencing files under the publicly accessible Mendeley Data repository. We have included a file for all source data DOIs for readers as an Expanded View Table (Table EV2). The table includes a description of each file, the corresponding figure information, and the DOI to access the data. This has also been reflected now in the Data Availability section of the manuscript. All other source data (excel files and uncropped protein blots) are included in the source data zip file with a README file.

Additionally, Figure 1b is a schematic and does not have data associated with it.

>> Dataset EV legends: Table EV1 should be renamed to Dataset EV1 with the legend inserted as a separate sheet and with appropriate callout. Table EV2 should be renamed to Table EV1 with the corresponding callout

The file names have been changed throughout the text.

>> Please consider additional changes and comments from our production team as indicated below:

- Figure Legends (main + EV):

"Please note that the figure legend style does not comply with the journal guidelines i.e. all the figure legends are in a run-on style."

The figure legend style has been changed to comply with journal guidelines.

"1. Please indicate the statistical test used for data analysis in the legends of figures 2b; 5a

The statistical pipeline and replicates of Figure 2B has been described. Fig 5a does not have statistical tests. The asterisks for Figure 5a are not indicative of p-values but serve as a label for amino acid residues. We have noted this in the figure legend to avoid confusion.

2. Please note that in figure 3e there is a mismatch between the annotated p values in the figure legend and the annotated p values in the figure file that should be corrected.

This figure legend has been changed to match the annotated p-values.

3. Please define the annotated p values **/* in the legends of figures 5a; 6c as appropriate."

Referee#1:

The authors satisfactorily addressed all points raised on the original version of the study. I fully support the publication of this interesting manuscript in EMBO Journal which elucidates the molecular interactions of ER and mitochondrial membranes in protein trafficking. It considerably advances our understanding of the cooperation and competition of different insertion systems for tail-anchored membrane proteins.

We thank the reviewer for their helpful comments and insightful feedback during this process.

Referee#2:

This is a revised manuscript describing the role of the endolysosomal system in regulating mitophagy receptor levels. Overall, the authors have thoroughly addressed the comments, and while open questions still remain, the study represents an important advance in not only our understanding of mitophagy regulation, but also the regulation of mitochondrial tail-anchored proteins. The study is likely to appeal to a broad audience and is clearly suitable for EMBO J.

Given that mysteries still remain regarding how the majority of BNIP3 is not turned over by mitophagy (i.e. potentially escaping mitophagosomes to the ER for endolysosomal turnover), it would be worthwhile to include a brief discussion of this open question as an area for future work. The authors may also wish to refer to precedent for this observation for other mitochondrial proteins during mitophagy (PMID: 23361001). For BNIP3, it would need to reside at mitochondria long enough to initiate mitophagy, before being shipped off to the ER for endolysosomal turnover.

This is an interesting discussion point which we have now included in lines 454-464. We have included the suggested reference (PMID: 23361001) as well as a related reference (PMID: 28381481) that also supports this model.

We thank the reviewer for their continued engagement with our manuscript and the many improvements that resulted from this dialogue.

Dear Christopher,

Thank you for submitting the revised version of your manuscript. I have now evaluated your amended manuscript and concluded that the remaining minor concerns have been sufficiently addressed.

Thus, I am pleased to inform you that your manuscript has been accepted for publication in the EMBO Journal.

Please note that it is The EMBO Journal policy for the transcript of the editorial process (containing referee reports and your response letter) to be published as an online supplement to each paper.

If you do NOT want the transparent process file published, you will need to inform the Editorial Office via email immediately. More information is available here: https://www.embopress.org/transparent-process#Review_Process

On a different note, I would like to alert you that EMBO Press offers a format for a video-synopsis of work published with us, which essentially is a short, author-generated film explaining the core findings in hand drawings, and, as we believe, can be very useful to increase visibility of the work. This has proven to offer a nice opportunity for exposure i.p. for the first author(s) of the study. Please see the following link for representative examples and their integration into the article web page:

<https://www.embopress.org/doi/full/10.15252/embj.2019103932>

Finally, we have noted that the submitted version of your article is also posted on the preprint platform bioRxiv. We would appreciate if you could alert bioRxiv on the acceptance of this manuscript at The EMBO Journal in order to allow for an update of the entry status. Thank you in advance!

If you have any questions, please do not hesitate to call or email the Editorial Office.

Best regards,

Daniel

Daniel Klimmeck, PhD
Senior Editor
The EMBO Journal
EMBO
Postfach 1022-40
Meyerohofstrasse 1
D-69117 Heidelberg

contact@embojournal.org
Submit at: <http://emboj.msubmit.net>

Rev_Com_number: RC-2023-01934
New_manu_number: EMBOJ-2023-114702R1
Corr_author: Shoemaker
Title: The EMC governs lysosomal turnover of a mitochondrial TA protein, BNIP3, to restrict mitophagy